# Aluminum foil negative electrodes with multiphase microstructure for all-solid-state Li-ion batteries

Yuhgene Liu[1], Congcheng Wang[2], Sun Geun Yoon[2], Sang Yun Han[2], John A. Lewis[1], Dhruv Prakash[1], Emily J. Klein[1], Timothy Chen [2], Dae Hoon Kang[3], Diptarka Majumdar[3], Rajesh Gopalaswamy [3] & Matthew T. McDowell [1,2] ✉

Metal negative electrodes that alloy with lithium have high theoretical charge storage capacity and are ideal candidates for developing high-energy rechargeable batteries. However, such electrode materials show limited reversibility in Li-ion batteries with standard non-aqueous liquid electrolyte solutions. To circumvent this issue, here we report the use of non-pre-lithiated aluminum-foil-based negative electrodes with engineered microstructures in an all-solid-state Li-ion cell configuration. When a 30-μm-thick $Al_{94.5}In_{5.5}$ negative electrode is combined with a $Li_6PS_5Cl$ solid-state electrolyte and a $LiNi_{0.6}Mn_{0.2}Co_{0.2}O_2$-based positive electrode, lab-scale cells deliver hundreds of stable cycles with practically relevant areal capacities at high current densities (6.5 mA cm$^{-2}$). We also demonstrate that the multiphase Al-In microstructure enables improved rate behavior and enhanced reversibility due to the distributed LiIn network within the aluminum matrix. These results demonstrate the possibility of improved all-solid-state batteries via metallurgical design of negative electrodes while simplifying manufacturing processes.

To meet the demands of long-range electric vehicles and electric flight, next-generation batteries must have higher energy density and improved safety. Solid-state batteries (SSBs) can potentially enable the use of new high-capacity electrode materials while avoiding flammable liquid electrolytes. Lithium metal negative electrodes have been extensively investigated for SSBs because of their low electrode potential and high theoretical capacity (3861 mAh g$^{-1}$)[1]. However, challenges associated with interfacial instabilities and lithium filament penetration to cause short-circuiting have proven extremely difficult to solve[1–4].

Materials that alloy with lithium at low potentials ("alloy negative electrodes") are an attractive alternative to lithium metal due to their high-lithium storage capacity and mitigation of filament growth[5]. Alloy-negative electrodes such as silicon have been investigated for decades for use in Li-ion batteries[6–9], and silicon is currently being incorporated in small fractions to boost the capacity of graphite-based

Li-ion battery-negative electrodes[10]. However, alloy-negative electrodes undergo substantial volumetric and structural changes during reaction with lithium[11], which causes excessive solid-electrolyte interphase (SEI) growth and accelerated cell failure within liquid electrolytes because of continual surface dimensional changes[12].

While silicon has attracted by far the greatest interest, other alloy-negative electrode materials also offer significant performance gains. One such candidate, aluminum, was first investigated as a lithium storage electrode in the 1970s[13,14]. The lithiation of aluminum to form the $\beta$-LiAl phase corresponds to a theoretical specific capacity of 990 mAh g$^{-1}$ and a volume change of 96%, lower than the 310% volume change of silicon[12]. Most importantly, aluminum is an abundant commodity metal that is cost-effectively manufactured as foils; direct use of foil-negative electrodes would boost cell energy density while also eliminating costs associated with conventional graphite slurry casting and solvent recycling[15,16]. Foils could also simultaneously act as the

[1]School of Materials Science and Engineering, Georgia Institute of Technology, Atlanta, GA 30332, USA. [2]George W. Woodruff School of Mechanical Engineering, Georgia Institute of Technology, Atlanta, GA 30332, USA. [3]Novelis, Inc., Kennesaw, GA 30144, USA. ✉e-mail: mattmcdowell@gatech.edu

active lithium storage medium and the current collector, further enhancing specific energy/energy density. However, aluminum-based foils have shown poor performance in batteries with non-aqueous electrolyte solutions under practically relevant conditions[17–20]. Degradation of aluminum electrodes is thought to occur due to porosity formation and SEI growth in liquid electrolytes[21–24], diffusional trapping of lithium[25–28], and mechanical fracture[14,29–33].

SSBs offer an entirely different chemo-mechanical environment compared to Li-ion batteries[5]. For instance, solid-state electrolytes (SSEs) do not flow to wet the surface of volume-changing negative electrode particles, which could stabilize SEI formation. Indeed, SSBs with silicon-based negative electrodes have recently been shown to exhibit improved cycling stability compared to batteries using non-aqueous electrolyte solution[34–38]. Furthermore, SSBs with a variety of alloy-based negative electrodes (both silicon and aluminum) could achieve high-energy density and specific energy (Fig. 1a, b), even approaching that of lithium metal SSBs with excess lithium (see Supplementary Note 1 for calculation details). However, most recent alloy-negative electrode SSB demonstrations have used cast particulate or composite electrodes, which are conceptually similar to conventional Li-ion battery electrodes. Given the different chemo-mechanical environment of SSBs, other electrode concepts may be viable for long-term durability, including the development of dense foil electrodes. Thick (>100 μm) indium or aluminum foils physically alloyed with lithium metal have been used as SSB negative electrodes to act as lithium sinks, but these thick foils have significant excess material and result in low-energy density that is unrealistic for practical use[16,39–41]. Furthermore, obviating the use of lithium metal for prelithiation is beneficial for scaled battery production.

Here, we demonstrate that SSBs with dense aluminum-based negative electrodes can exhibit stable electrochemical cycling using commercially relevant areal capacities (2–5 mAh cm$^{-2}$) and foil thicknesses (30 μm) without prelithiation. Aluminum and $Al_{94.5}In_{5.5}$ electrodes are incorporated within full-cell SSBs with argyrodite sulfide electrolyte ($Li_6PS_5Cl$), and the cells demonstrate hundreds of stable cycles at current densities up to 6.5 mA cm$^{-2}$ at 25 °C. Despite volume changes during cycling, the Al-based electrodes maintain their mechanical integrity without significant internal porosity formation, in contrast to their behavior in cells with non-aqueous electrolyte solutions, where failure occurs in ~70 cycles due to excessive SEI growth at internal pore surfaces. The addition of ~5 at. % indium to form a layered multiphase microstructure improves rate behavior, initial Coulombic efficiency, and attained capacity. These performance enhancements are due to the distributed lithiated indium phase promoting the (de) lithiation reactions of aluminum with minimal overpotential, as well as mitigation of lithium trapping by the high-lithium-diffusivity LiIn phase. These results demonstrate that Al-based negative electrodes could be realized within solid-state architectures and offer microstructural design guidelines for improved performance, potentially enabling high-energy-density batteries that avoid degradation challenges associated with lithium metal negative electrodes.

## Results and discussion

Two different types of negative electrode foils with 30-μm thickness were investigated herein: high-purity aluminum foil (99.999% aluminum) and an alloy with 5.5 at% indium. The 30-μm thickness of these foils corresponds to an areal capacity of ~8 mAh cm$^{-2}$ in the fully lithiated state; this thickness was selected because it can enable commercially relevant capacities (2–5 mAh cm$^{-2}$) while still retaining unreacted aluminum that can potentially be used as the current collector. Using a single foil as active material and current collector has previously been proposed to increase energy density[42]. Full cells were

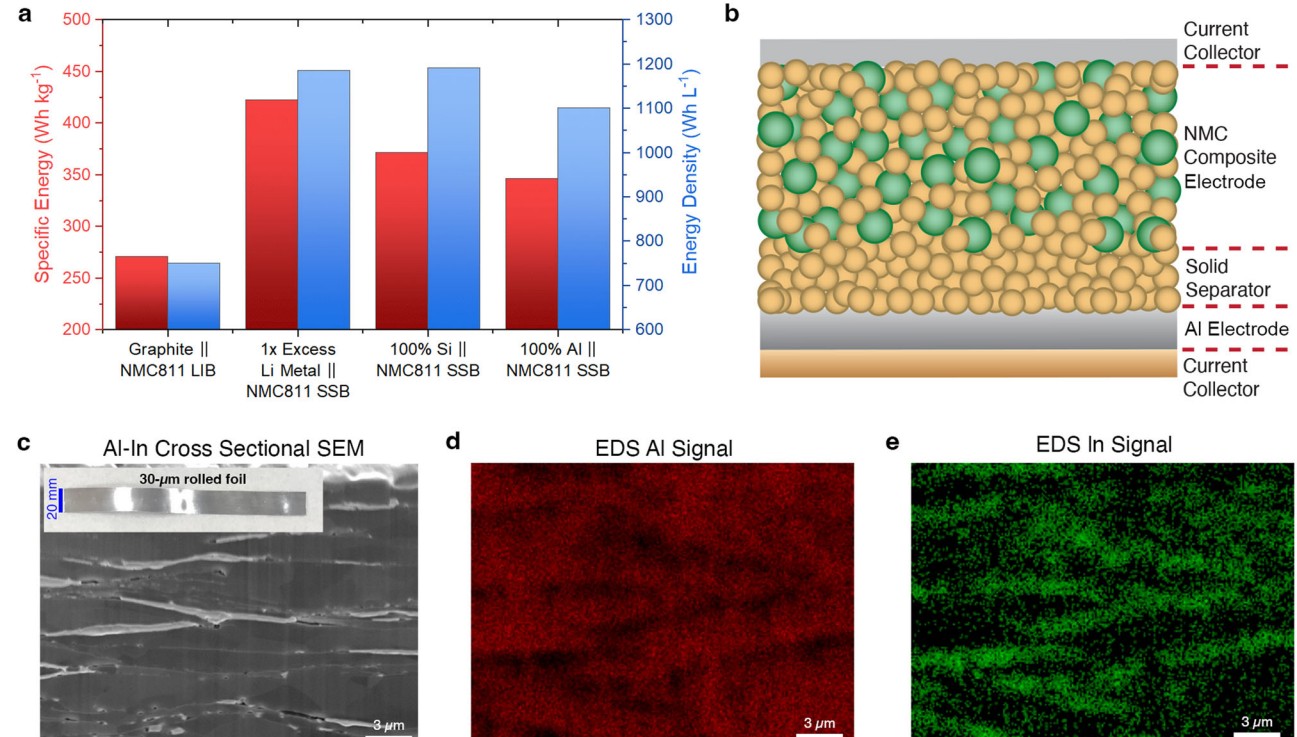

**Fig. 1 | Energy metrics of various negative electrodes within SSBs and structure of negative electrodes. a** Theoretical stack-level specific energy (Wh kg$^{-1}$) and energy density (Wh L$^{-1}$) comparison of a Li-ion battery (LIB) with a graphite composite negative electrode and liquid electrolyte, a SSB with 1× excess lithium metal at the negative electrode, a SSB with a dense silicon negative electrode, and a SSB with a dense aluminum negative electrode (see SI for details). **b** Schematic of a SSB with an aluminum-based negative electrode, SSE separator, and NMC composite positive electrode. The brown spheres represent the SSE and the green spheres represent NMC. **c** Cryo-FIB-SEM image of a pristine Al–In alloy foil; the lighter-contrast regions correspond to indium. Inset: photograph of a rolled foil. **d** EDS map of aluminum signal from a different SEM cross-section. **e** EDS map of indium signal.

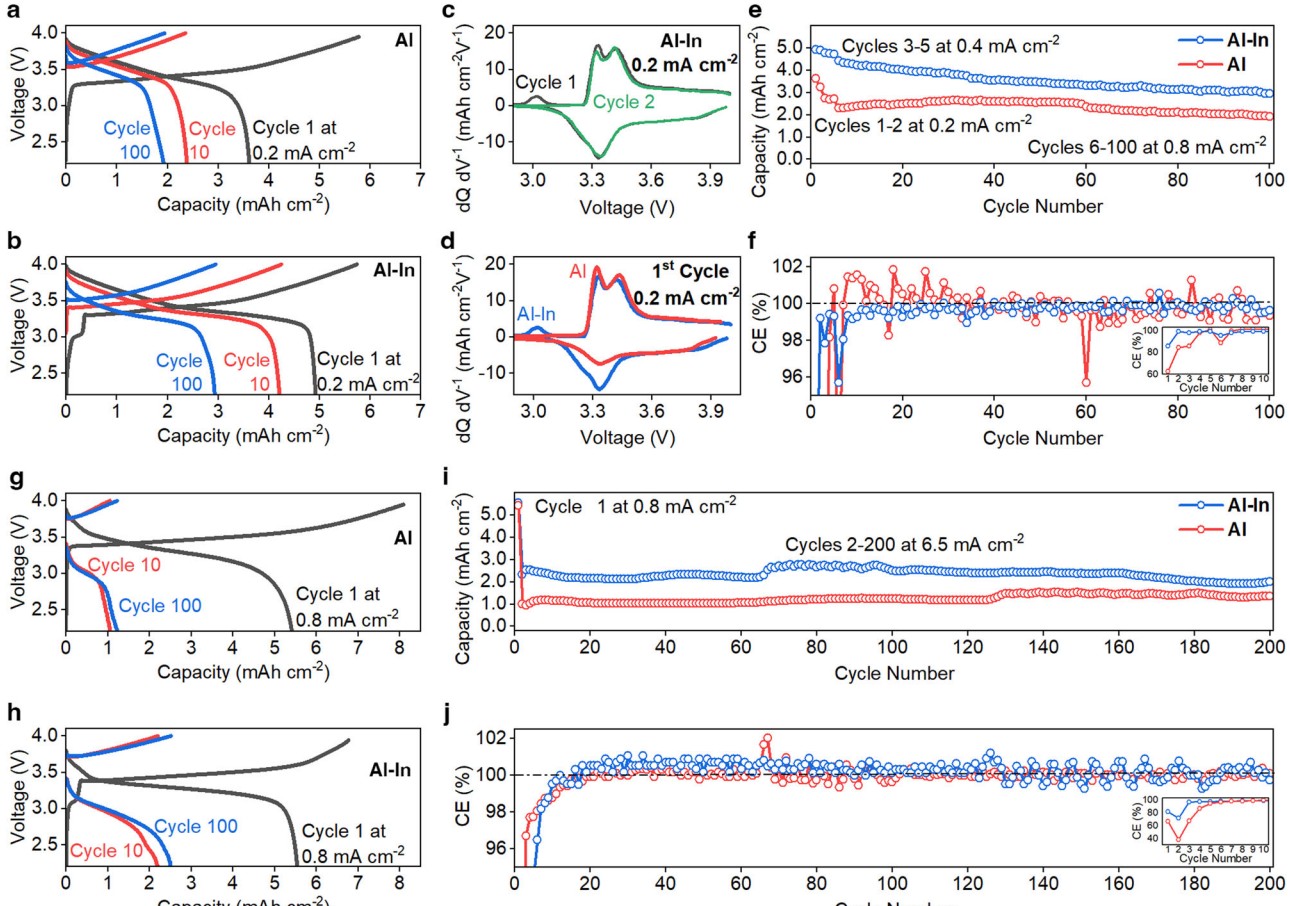

**Fig. 2 | Electrochemical behavior of all-solid-state cells with aluminum-based negative electrodes. a–f** Galvanostatic testing of aluminum and $Al_{94.5}In_{5.5}$ cells at 0.2 mA cm$^{-2}$ for the first two cycles, 0.4 mA cm$^{-2}$ for the next three cycles, and 0.8 mA cm$^{-2}$ for subsequent cycles (24 MPa stack pressure, 5.8 mAh cm$^{-2}$ positive electrode loading). **a** Aluminum|LPSC|NMC cell voltage curves. **b** $Al_{94.5}In_{5.5}$|LPSC|NMC cell voltage curves. **c** d$Q$/d$V$ curves for the first two cycles of the Al-In cell from (**b**). **d** d$Q$/d$V$ curves comparing the first cycle of the aluminum to the Al-In cell. **e** Areal capacity with cycling of both cells. **f** CE with cycling, with the inset showing CE over the first ten cycles. **g–j** Galvanostatic testing of aluminum and $Al_{94.5}In_{5.5}$ cells at higher current densities (0.8 mA cm$^{-2}$ for the first cycle and 6.5 mA cm$^{-2}$ for subsequent cycles, 50 MPa stack pressure, 8.3 mAh cm$^{-2}$ positive electrode loading). **g** Aluminum|LPSC|NMC cell voltage curves. **h** $Al_{94.5}In_{5.5}$|LPSC|NMC cell voltage curves. **i** Areal capacity with cycling. **j** CE with cycling, with the inset showing CE over the first ten cycles; the drop in CE on cycle 2 in the inset is due to the increased current density during this cycle. The increases in capacity in (**i**) at cycle 65 for $Al_{94.5}In_{5.5}$ and 125 for aluminum are due to slightly increased ambient temperature; all testing was otherwise performed at 25 °C.

assembled with $Li_6PS_5Cl$ (LPSC) as the SSE and $LiNb_{0.5}Ta_{0.5}O_3$-protected $LiNi_{0.6}Mn_{0.2}Co_{0.2}O_2$ (NMC622) as the active material within a composite positive electrode with 27.5 wt % LPSC (see "Methods").

Al–In alloy-negative electrodes were fabricated by melting appropriate ratios of each metal within an inert environment, then cooling and rolling to the desired thickness (Fig. 1c, inset). X-ray diffraction (XRD) revealed that the alloy was comprised of separate aluminum and indium phases (Supplementary Fig. 1), consistent with the solid-phase immiscibility during monotectic cooling from the Al–In phase diagram[43]. Figure 1c shows a cryogenic focused-ion beam (cryo-FIB) scanning electron microscopy (SEM) image of a pristine 30-µm-thick $Al_{94.5}In_{5.5}$ foil, and Fig. 1d, e shows X-ray energy-dispersive spectroscopy (EDS) analysis revealing elemental distribution in the material, where the mass ratio of indium to aluminum was verified to be ~1:4. The $Al_{94.5}In_{5.5}$ foil exhibits a distinctive laminar microstructure, with indium layers distributed throughout the aluminum matrix. The pure aluminum foils are also dense without cross-sectional morphological features.

Figure 2a, b shows galvanostatic cycling results from aluminum|LPSC|NMC622 and $Al_{94.5}In_{5.5}$|LPSC|NMC622 cells. These cells had positive electrode loadings of 5.8 mAh cm$^{-2}$, and they were held under 24 MPa stack pressure during cycling at 25 °C. Almost the entire

positive electrode capacity was utilized on the first charge at 0.2 mA cm$^{-2}$ in both cells, but the cell with the Al–In alloy showed a significantly higher initial Coulombic efficiency (CE) compared to the cell with the pure aluminum (85% vs 64%). The Al–In negative electrode cell showed an initial shoulder during charge associated with lithiation of indium (Fig. 2b). After the first cycle, the voltage curves in Fig. 2b and the differential capacity (d$Q$/d$V$) curves in Fig. 2c show little evidence for further lithiation of indium, indicating that the indium within the foil remains lithiated even after discharge. Differential capacity curves comparing the first cycle of two cells with different negative electrodes (Fig. 2d) highlight the improved reversibility of the Al–In-based cell.

After the first two cycles of these cells, the current density was increased to 0.4 mA cm$^{-2}$ for three cycles and then to 0.8 mA cm$^{-2}$ until the completion of 100 cycles, and Fig. 2e, f shows the corresponding areal capacity and CE. Cells with both negative electrodes exhibited good cycling stability with some decay under these conditions, with no evidence of short-circuiting. The $Al_{94.5}In_{5.5}$ negative electrode cell showed higher areal capacity than the pure aluminum cell (3–4 mAh cm$^{-2}$ vs. 2–3 mAh cm$^{-2}$, Fig. 2e). The CE values rapidly increased to >99% over the first few cycles, and the Al–In cell exhibited an average CE of 99.68% from cycle 5 through 100. The cell with the

pure aluminum negative electrode showed more erratic CE values with some over 100%, which is likely a result of trapped lithium within the material in the first few cycles[14]. The electrochemical performance and stability of the cell with the Al–In foil negative electrode approaches those of a cell with a pure indium foil negative electrode with a similar thickness (Supplementary Fig. 2), which exhibited an initial CE of 86% and stable cycling for hundreds of cycles. It is clear from these results that the inclusion of small amounts of indium within aluminum foils improves cycling capacity, CE, and stability.

We also tested cells with higher positive electrode loading and higher current densities at 25 °C to understand behavior under more aggressive cycling conditions. Figure 2g, h shows voltage curves from galvanostatic testing of aluminum | LPSC | NMC622 (Fig. 2g) and $Al_{94.5}In_{5.5}$ | LPSC | NMC622 (Fig. 2h) cells using a current density of 6.5 mA cm⁻², a positive electrode loading of 8.3 mAh cm⁻², and 50 MPa stack pressure. The first cycle for each cell was performed at the lower current density of 0.8 mA cm⁻². During first charge, almost the full volume of both foils was lithiated (~7–8 mAh cm⁻²), and the Al–In cell again showed higher initial CE (82%). The cells then exhibited stable cycling at the increased current density of 6.5 mA cm⁻² during cycles 2 through 200 (Fig. 2i, j). The Al–In cell showed particularly stable CE, with an average value of 99.98% from cycle 5 to 200. The lower areal capacities in Fig. 2i than in Fig. 2e are due to the much higher current density. This rate capability is notable when compared to SSBs using lithium metal negative electrodes, which often cannot sustain current

densities greater than a few milliamperes per cm² due to rapid filament growth and short-circuiting[44]; this demonstrates the distinct benefit of engineered alloy-negative electrodes over lithium metal for SSBs.

As an additional cycling test to examine longer-term durability, $Al_{94.5}In_{5.5}$ electrodes were cycled in cells with a significant excess of positive electrode material (~16 mAh cm⁻²) under capacity-limited conditions. This type of test minimizes the influence of any positive electrode degradation in the cell since there is excess positive electrode active material present. Figure 3a shows that this cell exhibited 500 cycles with steady capacity and no short-circuiting, where lithiation areal capacity was controlled to be 2.1 mAh cm⁻² per cycle at a current density of 2.0 mA cm⁻². Supplementary Fig. 3 contains the first, 100th, and 300th voltage curves from this experiment, showing consistent curve shape from the 100th to 300th cycles. Supplementary Table 1 compares the cycling results in Figs. 2, 3a to other recent demonstrations of alloy-negative electrode-based SSBs.

The cycling results of the aluminum and $Al_{94.5}In_{5.5}$ foils in SSBs demonstrate improved stability compared to electrochemical cycling of identical foils in coin cells using non-aqueous electrolyte solutions, which is likely due to enhanced interfacial stability and reduced SEI growth. Supplementary Fig. 4 shows galvanostatic cycling data from foils cycled with a controlled areal capacity of 2.0 mAh cm⁻² per cycle with a Li metal counter electrode using a typical carbonate-based electrolyte. Both types of foils failed in less than 70 cycles; this result is typical and is due to excessive SEI growth caused by internal pore

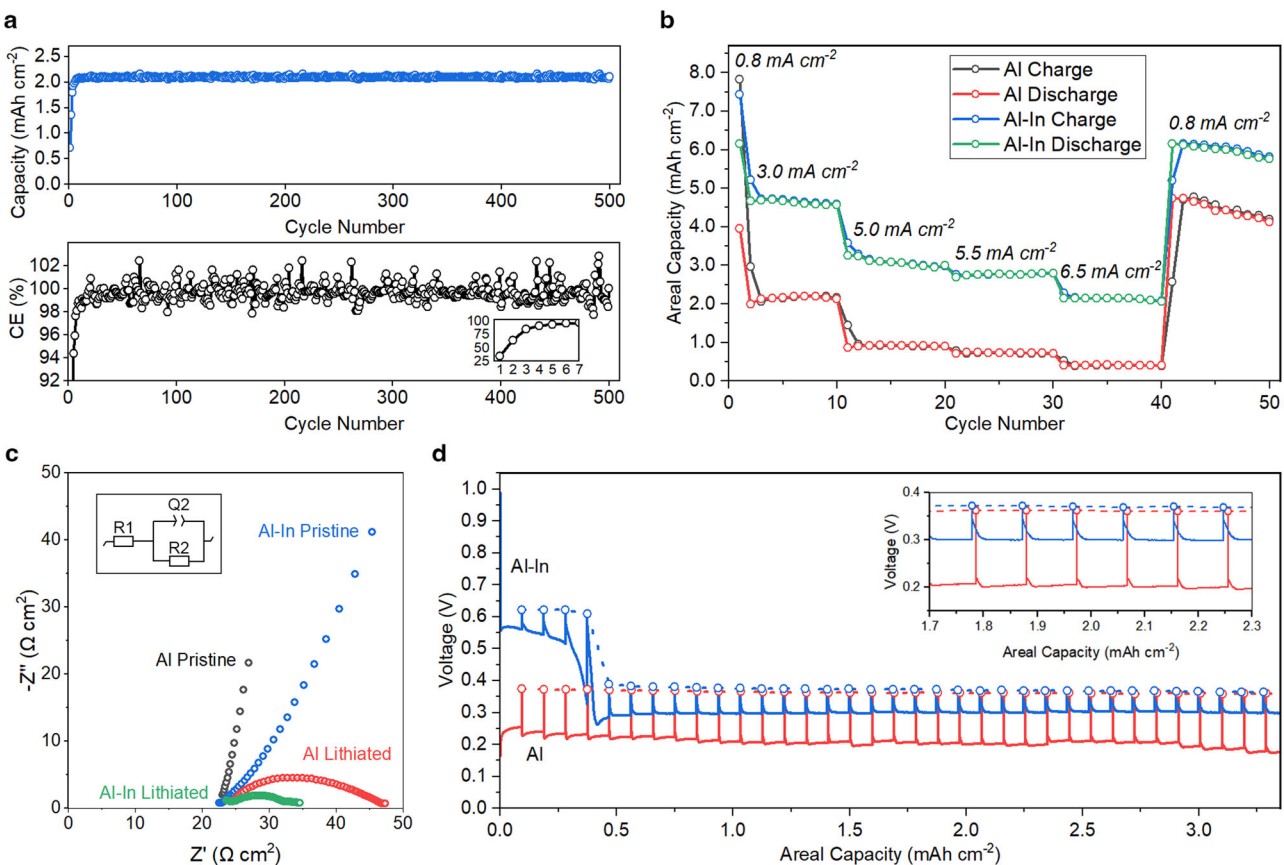

**Fig. 3 | Cycling, rate behavior, impedance, and GITT of aluminum-based electrodes in various cell configurations. a** Galvanostatic testing of an $Al_{94.5}In_{5.5}$ electrode at 0.5 mA cm⁻² for the first cycle and 2.0 mA cm⁻² for the subsequent cycles under constant-capacity testing conditions (lithiation capacity per cycle controlled to be 2.1 mAh cm⁻²). This cell has a significant excess of NMC at the positive electrode (16 mAh cm⁻²), and it was tested under 50 MPa stack pressure. **b** Rate testing of Aluminum | LPSC | NMC and $Al_{94.5}In_{5.5}$ | LPSC | NMC full cells with 8.3 mAh cm⁻² positive electrode loading, 50 MPa stack pressure, and current

densities denoted. **c** Nyquist plots and equivalent circuit of Aluminum | LPSC | NMC and $Al_{94.5}In_{5.5}$ | LPSC | NMC full cells with 8.3 mAh cm⁻² positive electrode loading and 50 MPa stack pressure. The equivalent circuit features two resistor elements ($R_1$ and $R_2$) as well as a constant phase element $Q_2$. **d** GITT measurements of Aluminum | LPSC | Li (red) and $Al_{94.5}In_{5.5}$ | LPSC | Li (blue) half cells with 10 MPa stack pressure. The open circles represent OCV values after the rest periods, and the solid lines are the voltage traces during current application. All testing was performed at 25 °C.

formation within the foils during alloying/dealloying[23]. We note that these coin cells were tested under lower stack pressures than the solid-state cells.

Figure 3b shows electrochemical rate testing experiments in which two SSB cells were subjected to increasing current densities, with the results demonstrating that the cell with the Al−In alloy shows better rate capability than the one with aluminum foil. Consistently higher areal capacities were achieved with the Al−In electrodes at current densities up to 6.5 mA cm$^{-2}$ under identical cell fabrication and testing conditions. Cycling of >2 mAh cm$^{-2}$ areal capacity at a current density of 6.5 mA cm$^{-2}$ indicates that the Al−In foil-based cell can exhibit relatively fast charge/discharge, although the cell is not optimized for fast rates.

Electrochemical impedance spectroscopy (EIS) measurements were carried out before and after charge of aluminum | LPSC | NMC622 and Al$_{94.5}$In$_{5.5}$ | LPSC | NMC622 cells (Fig. 3c). The spectra from cells with both types of negative electrodes in the pristine state show blocking behavior with extended Warburg tails, while depressed semicircles are present after charge. The charged cell with the Al−In negative electrode shows a higher-frequency depressed semicircle with a width of ~10.3 Ω cm$^2$ extracted via fit with the equivalent circuit shown in Fig. 3c, along with an additional low-frequency feature. The charged cell with the aluminum negative electrode shows a larger depressed semicircle with a width of ~23.5 Ω cm$^2$. Fitted parameters are shown in Supplementary Table 2. These data suggest that the presence of indium reduces the interfacial resistance of the negative electrode/ SSE interface.

To further investigate the influence of indium addition on electrochemical behavior, we used the galvanostatic intermittent titration technique (GITT). In this technique, current pulses are followed by rest periods, and the cell voltage during applied current and during rest periods can provide insight into kinetics and transport processes within the electrode[45,46]. Cells with Li metal counter electrodes were used to avoid any effects of composite positive electrodes in these experiments. Figure 3d shows GITT data for two cells with aluminum and Al$_{94.5}$In$_{5.5}$ working electrodes; a current of 0.4 mA was used for 10 min, followed by 10-h rest periods (Supplementary Fig. 5 shows typical relaxation data). The aluminum cell (red in Fig. 3d) shows slightly decreasing voltage over the current pulses, with the open-circuit voltage (OCV) relaxing to a constant value of ~0.36 V after each rest. The Al$_{94.5}$In$_{5.5}$ cell shows a higher plateau and OCV of ~0.62 V over the first ~0.37 mAh cm$^{-2}$, which corresponds to the lithiation of In to Li$_{x≤1}$In. The areal capacity for indium lithiation within the 30-μm-thick Al$_{94.5}$In$_{5.5}$ foil would be ~0.36 mAh cm$^{-2}$ assuming a theoretical capacity of 194 mAh g$^{-1}$ for the LiIn phase[16,47–49], which suggests that Li$_{0.9<x<1}$In forms during the first lithiation. After lithiation of indium, the voltage during the pulses then drops to a constant value of ~0.30 V during aluminum lithiation, with the OCV relaxing to ~0.37 V. These data show that the indium and aluminum are lithiated sequentially in accord with their distinct voltage plateaus despite being physically intermixed. Furthermore, the overpotential for the lithiation of aluminum in the Al−In electrode is ~100 mV lower than for the pure aluminum electrode, while the OCV values are very similar. This finding suggests faster kinetics during the lithiation reaction in the Al−In electrode.

Although GITT is sometimes used to extract diffusion coefficients from electrochemical data, this was not performed here since such analysis requires single-phase reaction behavior, and both the Al and In react via two-phase reactions[46]. However, prior work using nuclear magnetic resonance (NMR) techniques has directly measured Li diffusion coefficients in both the LiAl and LiIn phases[50]. The Li diffusion coefficients in both phases are quite high, with that for LiIn being approximately $10^{-6}$ cm$^2$ s$^{-1}$ and that for LiAl being approximately $10^{-7}$ cm$^2$ s$^{-1}$ at 25 °C. There is also some variation of the Li diffusion coefficient with composition in both phases since both have a narrow

range of single-phase solubility (for instance, from Li$_{48.3}$Al$_{51.7}$ to Li$_{53.1}$Al$_{56.9}$; see binary phase diagrams in Supplementary Fig. 6). The high Li diffusion coefficient in LiIn is consistent with the known high rate capabilities of pure In negative electrodes[51], and the lithiated phases of both In and Al can support relatively fast solid-state Li transport.

We next turn to the characterization of Al-based negative electrode evolution within SSBs to further understand the structural and morphological origins of the improved cycling performance. Ex situ X-ray diffraction (XRD) analysis was used to characterize the structural evolution of aluminum and Al$_{94.5}$In$_{5.5}$ foils throughout cycling. As shown in Fig. 4a, the pristine aluminum (ICDD 04-012-7848) was lithiated to form β-LiAl (ICDD 04-004-3791, see Supplementary Fig. 6), with some aluminum peaks remaining since this negative electrode was not fully lithiated (5.8 mAh cm$^{-2}$ charge transferred). After the first discharge (i.e., delithiation of the negative electrode), some of the aluminum peaks increase in intensity, but relatively weak β-LiAl peaks remain, corresponding to trapped lithium and consistent with the low initial CE for pure aluminum electrodes (Fig. 2). After 50 cycles in the delithiated (discharged) state, the XRD results show a mix of β-LiAl and aluminum with increased β-LiAl peak intensity, indicating increased retention of lithium in the electrode with cycling. XRD results for

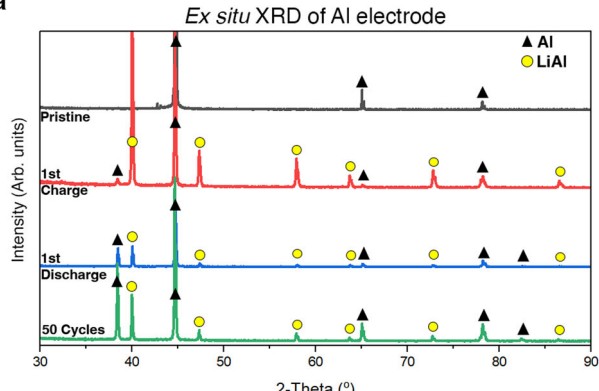

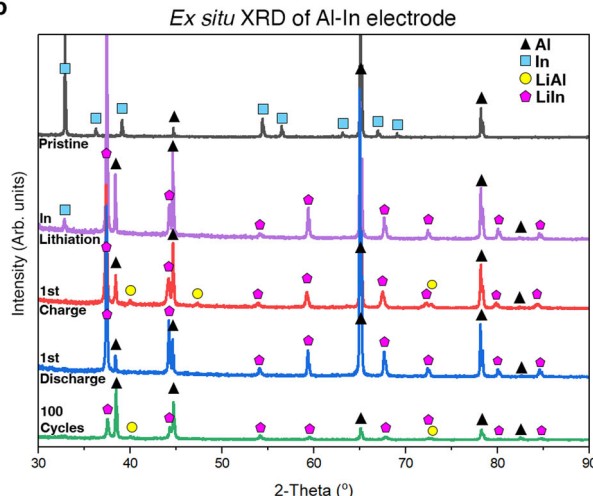

**Fig. 4 | Ex situ XRD characterization of aluminum-based negative electrodes before and after cycling. a** XRD analysis of aluminum negative electrodes in the pristine state (black), after initial full charge (red), after initial full discharge (blue), and after 50 cycles in the discharged state (green). **b** XRD analysis of Al$_{94.5}$In$_{5.5}$ negative electrodes in the pristine state (black), after indium lithiation (partial charge, purple), after full charge (red), after full discharge (blue), and after 100 cycles in the discharged state (green). All cells for (**a, b**) featured 5.8 mAh cm$^{-2}$ areal capacity in the positive electrode, lithiating ~70% of the Al foil and ~64% of the Al−In foil in the first charge; testing was performed at 25 °C.

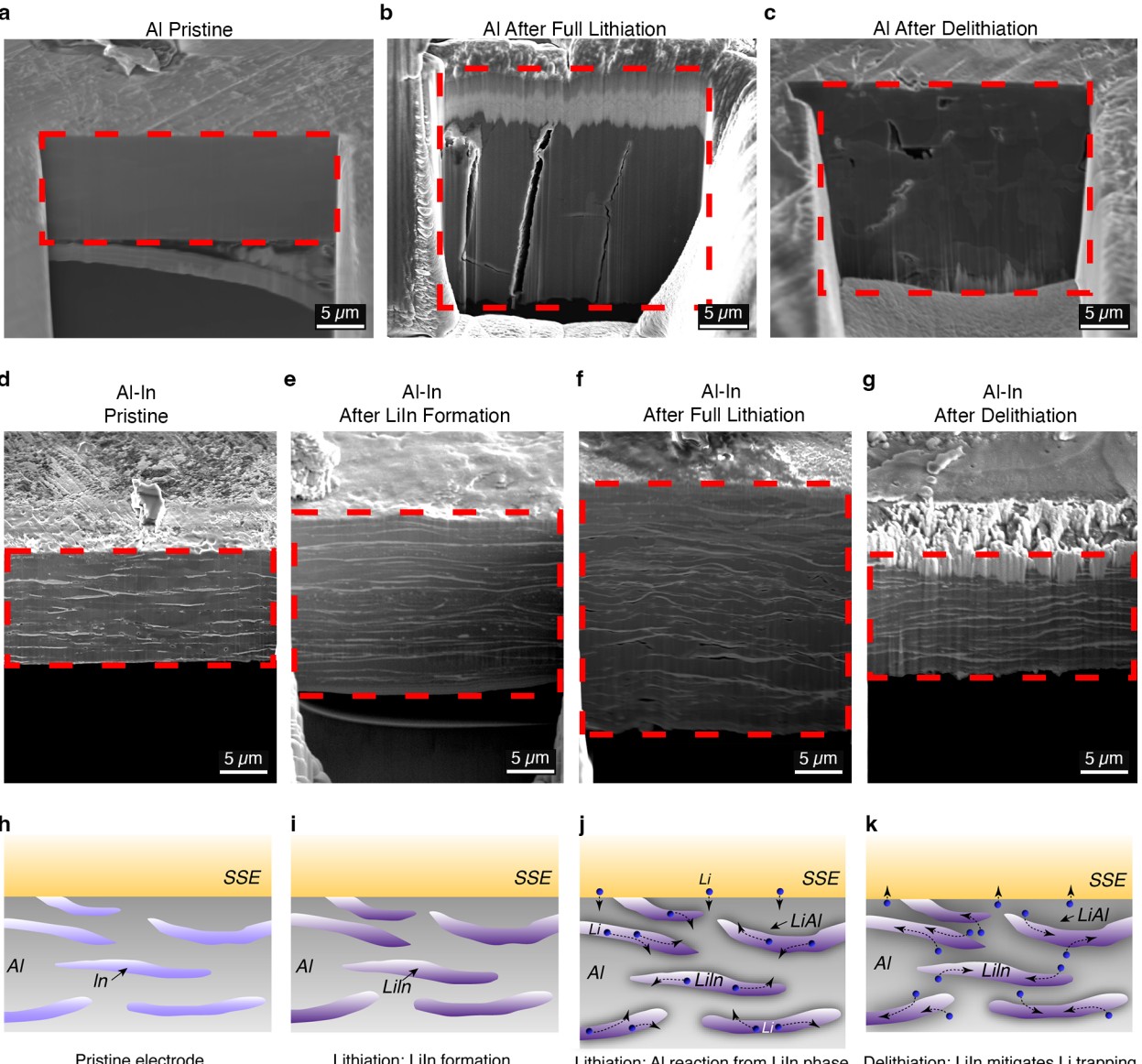

**Fig. 5 | Ex situ cryogenic FIB-SEM measurements of Al$_{94.5}$In$_{5.5}$ and aluminum foils at different stages of cycling in SSBs. a** Pristine aluminum, **b** aluminum after full lithiation, and **c** aluminum after delithiation. **d** Pristine Al$_{94.5}$In$_{5.5}$, **e** Al$_{94.5}$In$_{5.5}$ after LiIn formation, **f** Al$_{94.5}$In$_{5.5}$ after full lithiation, and **g** Al$_{94.5}$In$_{5.5}$ after delithiation. All cells for FIB-SEM were assembled with a stack pressure of 50 MPa and cycled at 0.7 mA cm$^{-2}$ with positive electrode loading of 3.0 mAh cm$^{-2}$, with both sets of foils being 11 μm in thickness instead of the 30 μm-thick foils used elsewhere to ensure FIB milling effectiveness. The red dotted lines in all images outline the cross-section. All testing was performed at 25 °C. **h–k** Schematics showing reaction mechanisms in the Al–In foil with a multiphase microstructure. **h** Pristine Al–In foil in contact with SSE. **i** The distributed indium phase is lithiated first to form LiIn. **j** The aluminum phase begins to react with lithium transported from the LiIn phase. **k** During delithiation, the LiIn phase remains lithiated and can transport lithium to avoid trapping by the pure aluminum phase.

Al$_{94.5}$In$_{5.5}$ foils are shown in Fig. 4b. The pristine foil shows a mix of aluminum and indium (ICDD 01-808-5363) phases. After partial charge so that only the indium is lithiated (purple trace), the spectrum shows the presence of LiIn (ICDD 04-017-5865, see Supplementary Fig. 6) and aluminum. After full charge (lithiation), both β-LiAl and LiIn diffraction peaks are evident, as well as aluminum peaks as above. Indium peaks are not present. After delithiation (discharge), only aluminum and LiIn peaks are present, without visible β-LiAl peaks. This is consistent with the relatively high CE of the Al$_{94.5}$In$_{5.5}$ electrodes in Fig. 2, indicating that most of the β-LiAl is delithiated but the LiIn is not. This result contrasts with the aluminum electrode in Fig. 4a, in which greater amounts of β-LiAl remained even after discharge. After 100 cycles in the discharged state (Fig. 4b), there remain peaks from Al, β-LiAl, and LiIn; indium peaks are not recovered. Together, these results show that the addition of small amounts of indium enables improved phase reversibility of the β-LiAl phase during cycling, which directly relates to improved electrochemical stability.

Figure 5 shows cryo-FIB-SEM imaging results from Al$_{94.5}$In$_{5.5}$ and pure aluminum electrodes at different stages of cycling. The cryogenic temperatures are useful for ensuring minimal sample damage and interaction with the Ga$^+$ milling beam. Figure 5a shows a pristine aluminum electrode. Figure 5b shows an aluminum electrode after full lithiation, where it has clearly grown in thickness and shows distinct cracks. Figure 5c shows an aluminum electrode after delithiation, and it contains some porosity. Supplementary Fig. 7 shows similar imaging results but with manual slicing of electrodes without FIB, with the delithiated aluminum showing increased porosity. Figure 5d shows a cryo-FIB image of a pristine Al$_{94.5}$In$_{5.5}$ foil, with the lighter-contrast indium phase visible as layers throughout the aluminum matrix. Figure 5e shows an Al$_{94.5}$In$_{5.5}$ foil electrode after lithiation of only the

indium. After full lithiation of the foil (Fig. 5f), the material undergoes a -100% thickness expansion, and after delithiation of the aluminum (Fig. 5g), the thickness of the foil has shrunk. Importantly, the LiIn layers remain intact and visible throughout the entire lithiation/delithiation process. A FIB-SEM image after 20 cycles (Supplementary Fig. 8) also shows intact and connected LiIn layers, indicating that the foil remains dense with interconnected LiIn layers throughout cycling. After undergoing 100 charge/discharge cycles, the foils remained mechanically intact (Supplementary Fig. 9).

Notably, these results diverge from testing of identical electrodes in cells using non-aqueous electrolyte solutions (Supplementary Fig. 10), where both types of foil materials grow from 30 to >200 μm in thickness at failure in less than 100 cycles. This thickening occurs because the foils become highly porous (Supplementary Fig. 10) due to the alloying/dealloying process, and the liquid electrolyte infiltrates the foil to cause continuous interior SEI growth[23]. The foils in SSBs exhibit less extensive SEI growth due to the planar interface with the SSE, and the higher stack pressure and all-solid nature of the SSB cell stack likely also assist in maintaining dense foils despite structural transformations.

Taken together, these findings provide evidence that the distributed LiIn phase within the aluminum matrix is important for enhancing the reversibility, rate behavior, and performance of Al–In electrodes. As already noted, the indium phase is lithiated first and stays lithiated even after discharge of Al–In||NMC622 cells, as shown in the schematic in Fig. 5h, i. This LiIn phase, which can support relatively fast Li diffusion, likely influences behavior in the following ways. First, since the LiIn phase is distributed throughout the aluminum matrix as a layered 3D network, there is a greater interfacial area available for the reaction of the aluminum with lithium from the LiIn phase. This enables transport of lithium from the LiIn network to react with aluminum to form LiAl with a lower overpotential (Fig. 5j). This idea is supported by the GITT measurements in Fig. 3d, where the Al–In electrode shows approximately -100 mV lower overpotential during aluminum lithiation compared to the pure aluminum electrode, while both show almost the same OCV values. The improved rate behavior in Fig. 3b is also likely due to this 3D distributed network effect. Second, the distributed LiIn phase appears to play an important role in minimizing lithium trapping, which is a known failure mechanism in aluminum electrodes since the pure delithiated aluminum phase exhibits a low Li diffusion coefficient and can act as a physical barrier to further Li extraction[52]. The distributed LiIn phase provides high-diffusivity transport channels through which Li can be removed through the surrounding delithiated aluminum phase (Fig. 5k), enabling the high initial CE observed in the Al–In||NMC622 cells (Fig. 2). Overall, these data show that the design concept of an interspersed mixed-ion-electron-conducting phase within a dense foil proves to be effective for improved CE and rate behavior.

We also investigated the effects of varying indium content on electrochemical performance. Samples were prepared with 0.2, 1.2, 2.5, 5.5, and 10 at% indium. All electrodes formed phase-separated layered microstructures (Supplementary Fig. 11). Electrochemical testing showed that the electrodes with less than 2.5 at% indium enabled diminished discharge capacity and CE, while the electrodes with 5.5 and 10 at% indium enabled the highest reversible capacity and CE. This result suggests that a minimum threshold of indium is needed to enhance performance.

Finally, we examined the effects of applied stack pressure on electrochemical behavior. For Al$_{94.5}$In$_{5.5}$|LPSC|NMC622 cells, stack pressures between 15 and 70 MPa showed similar first-cycle voltage curves and CE, but with slightly improved cycling stability at 50 MPa (Supplementary Fig. 12). Insignificant benefit was found beyond 50 MPa. This range of stack pressures is consistent with recent reports on alloy-negative electrodes for SSBs[34,35], and it is lower than other demonstrations with lithium metal negative electrodes[53]. We note that the stack pressure affects all components of these cells (positive electrode, separator, negative electrode, and interfaces), and that different stack pressures could possibly influence the morphological evolution of the electrode itself. Additional work is needed to understand the impact of stack pressure on alloy-negative electrodes and their interactions with the rest of the cell stack. Importantly, though, it is expected that SSB architectures could enable advantages for alloy-negative electrodes even at low stack pressures, since the SEI formation in the inner structure of the negative electrode is unlikely in solid-state cell configurations while it happens within the electrode's porous regions in cells with liquid non-aqueous electrolyte solutions[23], leading to rapid capacity decay (Supplementary Fig. 10).

Alloy-based negative electrodes have long been pursued for cells with non-aqueous electrolyte solutions but have not achieved stable cycling under practically relevant areal capacity and electrode thickness conditions. Our findings show the distinct benefits of solid-state architectures, as well as microstructure engineering of the negative electrode, for enabling stable all-solid-state secondary Li-based cells. We find that dense aluminum-based negative electrodes remain compact during lithiation and delithiation within SSBs and avoid the extensive SEI formation that plagues alloys in cells with non-aqueous electrolyte solutions, limiting performance. This behavior is likely due to the mechanical confinement induced by the all-solid stack, as well as the relatively stable and planar interfacial contact between the negative electrode and the SSE (as opposed to the steadily increasing interfacial area in cells using non-aqueous electrolyte solutions). SSB cycling performance can be improved through the addition of minor alloying elements; 5.5 at% indium is shown to enhance reversibility and improve rate behavior. This is due to the distributed high-diffusivity LiIn phase enabling reaction of lithium with aluminum over a large interfacial area to enhance rate behavior, while also minimizing lithium trapping during lithium removal. These findings suggest the possibility of using foil alloy-based metal electrodes for all-solid-state Li-based batteries, thus, avoiding the need for slurry coating, which makes up a relatively large portion of costs and energy requirements in battery manufacturing[54]. Furthermore, foil alloy-based metal electrodes offer the possibility of using one structure as both the ion-storage electrode and the current collector. Future efforts toward optimizing alloy composition and microstructure, determining the effects of other elemental additions beyond indium, and understanding material evolution are expected to enable further performance improvements.

## Methods
### Negative electrode preparation
99.999% 30-micron aluminum foil (Laurand Associates) was used as received for cell assembly and testing. Indium foil (99.995%, Sigma-Aldrich) was used to fabricate Al–In alloys. Stoichiometric amounts of aluminum and indium were placed in a MgO crucible (Sigma-Aldrich) within an argon-filled glove box (H$_2$O and O$_2$ content <1 ppm) and melted to 800 °C and held at that temperature for 1 h, followed by natural cooling. The ingots were then rolled via an electric roller (Durston) down to 30 μm.

### Positive electrode preparation
The positive electrode was a composite mixture of single-crystal LiNi$_{0.6}$Mn$_{0.2}$Co$_{0.2}$O$_2$ (NMC622) active material (MSE Supplies, particle size between 3 and 6 μm), Li$_6$PS$_5$Cl (MSE Supplies), and vapor grown carbon fiber (VGCF, Sigma-Aldrich). LiNb$_{0.5}$Ta$_{0.5}$O$_3$ (LNTO) was synthesized and coated on the active material to prevent side reactions with LPSC[55]. Stoichiometric amounts of niobium ethoxide (Sigma-Aldrich, 99.95%), tantalum butoxide (Sigma-Aldrich, 99.99%) and lithium acetate (Sigma-Aldrich, 99.95%) were dissolved in dry ethanol (Sigma-Aldrich, 99.5%) and stirred for 12 h. NMC622 powder was mixed into this solution via sonication in a Branson 1510 Ultrasonic Cleaner at 40 kHz for 2 h at 25 °C, and then the solvent was evaporated in a

vacuum oven. The dried powder was annealed in air at 450 °C for 1 min. The composition of the positive electrode was 70 wt.% of LNTO-coated NMC622, 27.5 wt.% LPSC, and 2.5 wt.% VGCF; this mixture was dry ball milled (Fritsch Pulverisette 7) at 150 rpm for 15 min three times in a $ZrO_2$ jar with eight $ZrO_2$ balls that was sealed in an argon-containing glove box ($H_2O$ <1 ppm and $O_2$ content <3 ppm) to create the positive electrode composite powder.

### All-solid-state battery assembly
$Li_6PS_5Cl$ (~1 μm particle size) from MSE Supplies was used as received for the SSE separator layer. 90 mg was uniaxially pressed at 125 MPa inside a 10-mm diameter polyether ether ketone (PEEK) die via titanium plungers (McMaster-Carr), forming a ~0.7-mm thick separator layer. The negative electrode (30-μm thickness) and appropriate amounts of the positive electrode composite powder were added before pressing the full cell to 375 MPa. After, 1 cm diameter graphite foil (GraFoil, McMaster) disks were added on opposite ends of the cell to ensure even pressure distribution across the cell. Then, the titanium plungers were reinserted to press the cell between the two graphite foil disks. Finally, the cell was placed under stack pressure via a custom pressure cell, which maintained an operating pressure by locking the cell between two steel plates with bolts at each corner[44]. The stack pressure was controlled to be between 15 and 70 MPa through precise tightening of the bolts with a digital torque wrench. The theoretical areal capacity of each cell (as determined by positive active material loading) is specified in figure captions, with 5.8 mAh cm$^{-2}$ corresponding to 31 mg cm$^{-2}$ NMC, 8.3 mAh cm$^{-2}$ corresponding to 44 mg cm$^{-2}$ NMC, and 3.0 mAh cm$^{-2}$ corresponding to 16 mg cm$^{-2}$ NMC.

### All-solid-state Li metal cell assembly
The separator layer was fabricated by pressing 90 mg of $Li_6PS_5Cl$ at 250 MPa inside the PEEK die via titanium plungers to create a ~0.7 mm LPSC layer. Then, the Al-based electrode working electrode was added before pressing to 375 MPa. 1 cm diameter graphite foil was added on top of the Al-based negative electrode, and a 1 cm diameter lithium metal disk (~13 mg and ~0.3-mm thick) was added to the opposite end onto the exposed solid electrolyte. Half cells were tested under a stack pressure of 10 MPa.

### Assembly of cells with lithium metal counter electrodes and non-aqueous electrolyte solutions
The electrochemical performance of cells with non-aqueous liquid electrolyte was evaluated using CR2032 coin cells. The Al and Al–In electrode foils were punched into disks with a diameter of 12 mm. Metallic lithium chips (MSE Supplies, 0.5-mm thick, 99.9% purity) were used as the counter electrode, and Celgard 2400 polymer films (25 μm thick, 41% porosity) were used as separators. 50 μL of electrolyte was used in each cell; the electrolyte composition was 1.0 M $LiPF_6$ in ethylene carbonate/diethyl carbonate (EC/DEC, 1:1 by volume, Sigma-Aldrich, battery grade) with 10 vol% fluoroethylene carbonate (FEC, Sigma-Aldrich, 99%). All coin cells were assembled in an argon-filled glove box ($H_2O$ <1 ppm and $O_2$ content <1 ppm). Once assembled, galvanostatic charge/discharge cycling tests were performed with a Landt Instruments battery cycler. The cells were tested with a voltage range of 0.01 to 1.0 V. For all cells with non-aqueous liquid electrolytes, the first two cycles of the galvanostatic charge-discharge testing featured a lower current density of 0.2 mA cm$^{-2}$ before cycling at 1 mA cm$^{-2}$. Electrochemical experiments were carried out at 25 °C without using an environmental chamber.

### Electrochemical characterization
Galvanostatic experiments were performed on a Landt Instruments battery cycler. All aluminum-based all-solid-state cells were charged to 4.0 V or to a time cutoff dependent on the theoretical amount of lithium that could be extracted from the positive electrode, and then

they were discharged to 2.2 V. GITT and potentiostatic electrochemical impedance spectroscopy (PEIS) measurements were performed with a Bio-Logic SP200 potentiostat. EIS was performed using a quasi-static potential mode with a voltage amplitude of 10 mV. Measurements were taken using a frequency range of 2 MHz to 2 Hz with 10 points per decade. The lithiated samples (red and green points in Fig. 3c) were lithiated under constant a current of 0.8 mA cm$^{-2}$, and the cells were held at open circuit for 30 s before the EIS measurements. All electrochemical experiments on solid-state battery cells were carried out at 25 °C in an argon-filled glove box ($H_2O$ <1 ppm and $O_2$ content <3 ppm) without using an environmental chamber.

### Materials characterization
The cycling cells were stopped and disconnected, and then they were disassembled inside an argon-filled glove box ($H_2O$ <1 ppm and $O_2$ content <3 ppm). The negative electrode was collected via delamination from the cell stack. For cross-sectional SEM-EDS, the foils were cut with a scalpel inside an argon environment before they were exposed to atmospheric conditions for ~40 s for sample transfer into the vacuum chamber. Images were collected on a Hitachi SU8230 SEM using an accelerating voltage of 15 kV and an 8 mm working distance. EDS was performed with an X-Max$^N$ X-ray detector (Oxford Instruments) under the same accelerating voltage and working distance. Aztec 2.3 software was used for elemental mapping. Cryogenic FIB-SEM images were collected on a Thermo Fisher Helios 5CX instrument using a temperature of −145 °C for milling, polishing, and imaging. FIB milling of the electrodes used 30 keV Ga$^+$ ions at 9.4 nA and polishing was performed at 2.5 nA. For XRD samples, the foils were placed between Kapton tape and a glass slide within a glove box ($H_2O$ <1 ppm and $O_2$ content <3 ppm). XRD data was collected on a Panalytical Empyrean instrument, with scans from 30 to 90° with a Cu-negative electrode as the X-ray source, 45 keV tension, 50 mA current, and copper K-α radiation.

## Data availability
Source data are provided with this paper.

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

## Acknowledgements

Support is acknowledged from Novelis, Inc. M.T.M. acknowledges support from a Sloan Research Fellowship in Chemistry from the Alfred P. Sloan Foundation. This work was performed in part at the Georgia Tech Institute for Electronics and Nanotechnology, a member of the National Nanotechnology Coordinated Infrastructure (NNCI), which is supported by the National Science Foundation (ECCS-2025462).

## Author contributions

Y.L. and M.T.M. conceived the ideas and designed the experiments. Y.L., C.W., S.H., J.L., and E.K. contributed to the methods, and Y.L., C.W., T.C., and S.Y. were involved with the investigation. Y.L. and C.W. conducted the formal analysis, and Y.L. and D.P. conducted the validation. Funding was acquired by M.T.M. M.T.M. provided supervision, and D.K., D.M., R.G., and M.T.M. were project administrators. Y.L. and M.T.M. co-wrote the manuscript, and all authors reviewed and edited the manuscript.

## Competing interests

Some of the authors are inventors on patent application PCT/US2023/017867 and provisional patent application 63/488,847 related to aluminum-based materials for solid-state batteries. The remaining authors declare no competing interests.
