## [Peer Review File · Nature Communications]

REVIEWER COMMENTS

Reviewer #1 (Remarks to the Author):

Firstly, there is much to appreciate about this work. The authors have assembled a highly functional solid state cell which shows impressive cycling stability at >200 cycles. This alone is commendable and a clear indication that the team's methods and design are quite robust.

The expectation that the addition of In to the anode might improve the cycling behavior should be very familiar to this group, as they previously have noted themselves: "Prior work on aluminum-based anodes has focused on modifying the composition to improve cyclability.¹¹ Combining aluminum with other elements that are inert or that actively alloy with lithium, such as zinc, tin, or silicon,¹⁴ has been shown to be beneficial." (<https://chemistry-europe.onlinelibrary.wiley.com/doi/10.1002/batt.202200363>)

However, the interpretation **why** the cells function as well as they do is not convincing and needs significant revision. To say plainly: The GITT method for interpreting the transport character of Li is simply not valid here. GITT is appropriately used for a single phase in a diffusion limited regime. In the case of the anode in question, The Al-LiAl system is two phase, and the addition of In, presumably leading to LiIn formation, could even lead to a three phase scenario with coexistence of Al, LiAl, and LiIn. The authors are strongly recommended to read the work by Zhu and Wang on this topic: <https://pubs.acs.org/doi/10.1021/jp9113333> This may give some ideas for how to conduct future experiments, so as to truly identify the nature of Li transport and the origin of the improved performance.

Li transport in LiIn may very well be faster than in LiAl, but the authors should rationalize this a little more in terms of the small amount of In in the anode and presumably good transport at room temperature through LiAl. A helpful reference may be found by Tarczón et al from 1987: <https://www.sciencedirect.com/science/article/pii/092150938890055X>. With all samples converging to diffusivities of 10^{-11} , this is orders of magnitude different compared to what is expected and what may be optimal.

In the case of these Al based electrodes, the use of GITT is further questionable since Ref[28] notes that the phase transformations of $\text{Li}+\text{Al} \rightarrow \text{LiAl}$ is reaction controlled. This further calls into question the role of diffusivity itself in the governance, and hence improvement of the Al and AlIn electrodes used here. Based on the cross sectional images by the authors in Fig 1c/1e, many reaction interfaces would be expected, particularly for the first few cycles. This same work also showed that less utilization of Al leads to longer electrode lifetime, with the substitution of In for Al, could this be the case in the submitted work as well?

Additionally, the authors make it a central highlight of the paper that the SS cells outperform their liquid counterparts, but to what extent is the stack pressure responsible for this change in performance? Perhaps this is an open question, but stack pressures of 25 MPa and 50MPa already surpass the yield strength of pure Al, and of course, In itself is incredibly soft. Could such stack pressure lead to discrepancies in anode evolution (e.g. porosity), which in turn has implications itself for Li transport (vis-a-vis tortuosity) and interfacial arrangements.

Lastly, given the visibility of Nature Communications and the complexity of the material evolution, the authors are requested to prepare an appropriate schematic illustration for future readers. For example, the NatComm paper published by Li et al (<https://www.nature.com/articles/s41467-020-15452-0>) has excellent illustrations to guide readers through their findings and methods.

The manuscript itself is well written, and the importance of the findings are also appropriate for NatComm. However, taking in mind the above comments, the reviewer suggests a Major Revision for the manuscript prior to acceptance/publication.

Reviewer #2 (Remarks to the Author):

In this manuscript, the authors designed a solid-state battery with Al-In foil anodes and enhanced the rate behaviors as well as reversibility. However, the logic of the article is somewhat confusing. In addition, the authors need to prove the storage mechanism of the materials through other characterization methods and theoretical calculations. Therefore, I'm afraid that this work cannot be considered for publishing on Nature Communications. Other comments are as follows:

1. Why the incorporation of small amounts of indium within aluminum foils can greatly influence the lithium diffusion kinetics? What's the synergetic effect between indium and aluminum?

2. The effects of Indium content and stack pressure should be considered first. In addition, the stack pressure should be consistent in the manuscript. For example, the stack pressure in Figure 2A-D and Figure S2 is 24 MPa, but it changed to 50 MPa in Figure 2(G, H). Moreover, the assembled stack pressure of the cells for SEM in Figure 5 changed to 24 MPa again.
3. Binary phase diagrams of Li-Al and Li-In may help to get further understanding of the alloy reaction.
4. In Ex situ X-ray diffraction tests, more decent steps during the lithiation process are needed to testify the alloy reaction orders of Li with Al and In.
5. What's the volume expansion ratio of pure In foil in lithiation process? Why the presence of In metal can reduce the volume expansion of the electrode?
6. Compared with the previously reported works, whether the rate performance, cycling stability and diffusion coefficients of the batteries are improved?
7. In Figure 3B, what does the Q2 represent in the equivalent circuit? It needs to be explained.
8. The storage mechanism is not clear. TEM characterization and other in-situ analysis need to be supplemented.
9. In the manuscript, the loading of the cathode (8.3 mAh/cm² or 5.8 mAh/cm²) are suggested to be fixed.
10. There are some errors should be corrected. For example, "(Fig. 1D)" in Line 4, Page 7; "2.5 2t.% VGCF" in Page 17; etc.
11. A comparison of recently reported hosts for Li storage in solid-state batteries should be provided, so that the readers can have a clear comparison between this material and other materials.

Reviewer #3 (Remarks to the Author):

Major revision: Suitable for Nature Communications after major revision.

In this work, the authors performed a study to demonstrate the benefits of solid-state architectures along with microstructure engineering of aluminum-based foils as anodes for rechargeable battery systems. Authors found that minor alloying of 5.5 at. % indium with aluminum led to enhance reversibility and improve rate behavior. Such aluminum-indium based foil anodes remained compact during lithiation and delithiation within solid state battery framework and avoided extensive SEI formation. Given the fact that the work is timely and thoroughly done, I would recommend this topic of

discussion to be published. However, there are few possible sources of ambiguity (as listed below) which need to be majorly revised before the decision of acceptance.

1. What is reason behind huge fluctuation of the CE versus cycle number for Al anode (as shown in Fig. 2F)? In contrast to Fig. 2F, Fig. 2I exhibits similar behavior for both the Al and Al-In anodes at higher current density. How do the authors justify this?
2. What is the origin of the drop (minimum at the cycle number 2) in the CE versus cycle plot, which is followed by a gradual increase, as shown Fig. 2J (inset)?
3. Line no. 211-212: What do the author mean by interfacial resistance? Do they mean the anode-electrolyte interface or the Al-In interface? Authors should clarify this point.
4. High order of Li-diffusivity for Al-In alloy anode is only observed at a controlled and low value of areal capacity (0.5 mAh cm^{-2}), as shown in Fig. 3D. In addition, the data points are fluctuating. Especially, the Li-diffusion behavior of both the Al and Al-In anodes become similar at the Al-plateau region. Hence, authors should explain the origin of the improved rate performance for Al-In anode with deeper insight.
5. A minor comment: (line no. 147) "Differential capacity curves comparing the first cycle of two cells with different anodes (Fig. 1D) highlight the improved reversibility of the Al-In-based cell." Numbering of figure is inappropriate.

Response to Reviewer Comments

“Aluminum Foil Anodes with Multiphase Microstructure for Solid-State Batteries”

Yuhgene Liu¹, Congcheng Wang², Sun Geun Yoon², Sang Yun Han², John A. Lewis¹, Dhruv Prakash¹, Emily J. Klein¹, Timothy Chen², Dae Hoon Kang³, Diptarka Majumdar³, Rajesh Gopalaswamy³, Matthew T. McDowell^{1,2*}

We thank the reviewers for their time and effort in reviewing our manuscript. In this response document, we list the reviewers' comments and describe the changes that have been made to the manuscript and SI in response to the comments. For clarity, the reviewer's comments are in black text, and our responses are in blue text. Additions and revisions to the manuscript and SI have been included in this response, and they are given in red text. Copies of the manuscript and SI in which changes are highlighted have also been included with this submission.

REVIEWER COMMENTS

Reviewer #1 (Remarks to the Author):

Comment: Firstly, there is much to appreciate about this work. The authors have assembled a highly functional solid state cell which shows impressive cycling stability at >200 cycles. This alone is commendable and a clear indication that the team's methods and design are quite robust.

Response: We thank the reviewer for their positive response.

Comment: The expectation that the addition of In to the anode might improve the cycling behavior should be very familiar to this group, as they previously have noted themselves: "Prior work on aluminum-based anodes has focused on modifying the composition to improve cyclability.¹¹ Combining aluminum with other elements that are inert or that actively alloy with lithium, such as zinc, tin, or silicon,¹⁴ has been shown to be beneficial." (<https://chemistry-europe.onlinelibrary.wiley.com/doi/10.1002/batt.202200363>)

Response: Indeed, the statement above is in our prior work on aluminum foils for liquid-cell lithium-ion batteries, and it refers to other recent work that has developed alloy compositions and tested them in liquid cells. As we show in our recent paper on liquid cells (the reference linked in the comment above), aluminum foils in liquid cells are fundamentally limited by excessive SEI that forms during cycling. While some studies have shown improved behavior due to alloying (e.g., S. S. Sharma, P. J. Crowley, A. Manthiram, *ACS Sustainable Chem. Eng.* 2021, 9, 14515), improvements have been modest compared to the results we present in the current manuscript on solid-state batteries.

Comment: However, the interpretation *why* the cells function as well as they do is not convincing and needs significant revision. To say plainly: The GITT method for interpreting the transport character of Li is simply not valid here. GITT is appropriately used for a single phase in a diffusion limited regime. In the case of the anode in question, The Al-LiAl system is two phase, and the addition of In, presumably leading to LiIn formation, could even lead to a three phase scenario with coexistence of Al, LiAl, and LiIn. The authors are strongly recommended to read the work by Zhu and Wang on this topic: <https://pubs.acs.org/doi/10.1021/jp9113333> This may give some ideas for how to conduct future experiments, so as to truly identify the nature of Li transport and the origin of the improved performance.

Li transport in LiIn may very well be faster than in LiAl, but the authors should rationalize this a little more in terms of the small amount of In in the anode and presumably good transport at room temperature through LiAl. A helpful reference may be found by Tarczon et al from 1987: <https://www.sciencedirect.com/science/article/pii/092150938890055X>. With all samples converging to diffusivities of 10^{-11} , this is orders of magnitude different compared to what is expected and what may be optimal.

In the case of these Al based electrodes, the use of GITT is further questionable since Ref[28] notes that the phase transformations of $\text{Li}+\text{Al}\rightarrow\text{LiAl}$ is reaction controlled. This further calls into question the role of diffusivity itself in the governance, and hence improvement of the Al and AlIn electrodes used here. Based on the cross sectional images by the authors in Fig 1c/1e, many reaction interfaces would be expected, particularly for the first few cycles. This same work also showed that less utilization of Al leads to longer electrode lifetime, with the substitution of In for Al, could this be the case in the submitted work as well?

Response: This is a key point, and we sincerely thank the reviewer for their thoughtful comments here. We have used this comment as a catalyst to improve our analysis in the revised manuscript, and in particular the reference from Tarczon (of which we were unaware, unfortunately) is important for this analysis – again, thanks to the reviewer for spending their time on this.

In short, we agree with the reviewer that the GITT analysis to extract diffusion coefficients was inappropriately applied in the previous version of the manuscript. Our original thought was that since the LiAl phase does exhibit single-phase Li solubility over a small range, relaxation within that phase could yield diffusion coefficients. However, after further consideration of the complicating factors as well as prior literature, we agree with the reviewer that the extracted diffusion coefficients in Fig. 3d of the prior version of the manuscript were inaccurate. However, we feel that the GITT experiment itself still contains valuable information.

To address this comment, we have revised the manuscript by removing the calculations of diffusivity from the paper, and instead rely on the GITT measurement, new characterization (detailed subsequently), and prior work to draw mechanistic conclusions. The changes we have made sharpen our conclusions, and we have listed them here.

On page 12 of the manuscript, we have removed our prior paragraphs on the extraction of lithium diffusion coefficients from the GITT data and have replaced it with the following paragraph:

“Although GITT is sometimes used to extract diffusion coefficients from electrochemical data, this was not performed here since such analysis requires single-phase reaction behavior, and both the Al and In react via two-phase reactions⁴⁴. However, prior work using nuclear magnetic resonance (NMR) techniques has directly measured Li diffusion coefficients in both the LiAl and LiIn phases⁴⁸. The Li diffusion coefficients in both phases are quite high, with that for LiIn being approximately $10^{-6} \text{ cm}^2 \text{ s}^{-1}$ and that for LiAl being approximately $10^{-7} \text{ cm}^2 \text{ s}^{-1}$ at room temperature. There is also some variation of the Li diffusion coefficient with composition in both phases since both have a narrow range of single-phase solubility (for instance, from $\text{Li}_{48.3}\text{Al}_{51.7}$ to $\text{Li}_{53.1}\text{Al}_{56.9}$; see binary phase diagrams in Supplementary Fig. 6). The high Li diffusion coefficient in LiIn is consistent with the known high rate capabilities of pure In anodes⁴⁹, and the lithiated phases of both In and Al can support relatively fast solid-state Li transport.”

On page 17 of the manuscript, we have rewritten the prior paragraph on proposed mechanisms as follows:

“Taken together, these findings provide evidence that the distributed LiIn phase within the aluminum matrix is important for enhancing the reversibility, rate behavior, and performance of Al-In electrodes. As already noted, the indium phase is lithiated first and stays lithiated even after discharge of full cells, as shown in the schematic in Fig. 5H(i-ii). This LiIn phase, which can support relatively fast Li diffusion, likely influences behavior in the following ways. First, since the LiIn phase is distributed throughout the aluminum matrix as a layered 3D network, there is a greater interfacial area available for the reaction of the aluminum with lithium from the LiIn phase. This enables transport of lithium from the LiIn network to react with aluminum to form LiAl with a lower overpotential (Fig. 5H(iii)). This idea is supported by the GITT measurements in Fig. 3D, where the Al-In electrode shows approximately $\sim 100 \text{ mV}$ lower overpotential during aluminum lithiation compared to the pure aluminum electrode, while both show almost the same OCV values. The improved rate behavior in Fig. 3B is also likely due to this 3D distributed network effect. Second, the distributed LiIn phase appears to play an important role in minimizing lithium trapping, which is a known failure mechanism in aluminum electrodes since the pure delithiated aluminum phase exhibits a low Li diffusion coefficient and can act as a physical barrier to further Li extraction⁵³. The distributed LiIn phase provides high-diffusivity transport channels through which Li can be removed through the surrounding delithiated aluminum phase (Fig. 5H(iv)), enabling the high initial CE observed in the Al-In cells (Fig. 2). Overall, these data show that the design concept of an interspersed mixed-ion-electron-conducting phase within a dense foil proves to be effective for improved CE and rate behavior.”

Note that the references to the schematic above (Fig. 5H) are discussed in response to a later comment by the reviewer.

On page 5 of the manuscript, we clarify the mechanisms: “These performance enhancements are due to the distributed lithiated indium phase promoting the (de)lithiation reactions of aluminum with minimal overpotential, as well as mitigation of lithium trapping by the high-lithium-diffusivity LiIn phase.”

On page 20: “This is due to the distributed high-diffusivity LiIn phase enabling reaction of lithium with aluminum over a large interfacial area to enhance rate behavior, while also minimizing lithium trapping during lithium removal.”

In the abstract: “The multiphase Al-In microstructure enables improved rate behavior and enhanced reversibility due to the distributed LiIn network within the aluminum matrix.”

Here is a list of further changes to figures and the SI:

- Removed the prior Fig. 3d, which contained diffusion coefficient data
- Removed Supplementary Figure 4b, which was focused on diffusion coefficient extraction
- Removed the paragraph in SI focused on diffusion coefficient extraction

To summarize these changes, we believe that the diffusivity of Li in the LiIn phase itself is not the key reason for improved behavior, although it is relatively high. Instead, we believe that the fact that the LiIn phase is distributed throughout the Al matrix is the key, as this creates 3D interfaces to minimize transport lengths for Li-Al reaction while also mitigating lithium trapping effects. Our revisions reflect these ideas.

Comment: Additionally, the authors make it a central highlight of the paper that the SS cells outperform their liquid counterparts, but to what extent is the stack pressure responsible for this change in performance? Perhaps this is an open question, but stack pressures of 25 MPa and 50MPa already surpass the yield strength of pure Al, and of course, In itself is incredibly soft. Could such stack pressure lead to discrepancies in anode evolution (e.g. porosity), which in turn has implications itself for Li transport (vis-a-vis tortuosity) and interfacial arrangements.

Response: This is an important point. The liquid cells reported in the paper are indeed tested in coin cells with lower stack pressure (~a few MPa). It is possible that foils in liquid cells under higher stack pressure could exhibit improved cyclability if the foils remain dense, although we believe this is somewhat outside the scope of our current study. However, a key point is that if morphological evolution of the foils in SSB cells is different at lower stack pressure, there will likely not be excessive SEI formation on this excess surface area as there is in liquid cells, since SSEs do not flow like liquids. Thus, we do expect that there are major advantages to the SSB architecture for these types of materials. Future work is intended to examine interfacial limitations that need to be overcome to enable lower stack pressures.

To address this comment, we have made the following changes to the manuscript.

On page 10, we have clarified that the liquid cell experiments were performed in coin cells at lower stack pressure than the solid-state experiments: “We note that these coin cells were tested under lower stack pressures than the solid-state cells.”

On page 19, we have made the following changes: “We note that the stack pressure affects all components of these cells (cathode, separator, anode, and interfaces), and that different stack pressures could possibly influence the morphological evolution of the foil itself. Additional work is needed to understand the impact of stack pressure on foil anodes and their interactions with the rest of the cell stack. Importantly, though, it is expected that SSB architectures could enable advantages for alloy foil anodes even at low stack pressures, since there will never be SEI formation in the interior of the foil as occurs at porous regions of foils in liquid cells²², leading to rapid capacity decay (Supplementary Fig. 10).”

Comment: Lastly, given the visibility of Nature Communications and the complexity of the material evolution, the authors are requested to prepare an appropriate schematic illustration for future readers. For example, the NatComm paper published by Li et al (<https://www.nature.com/articles/s41467-020-15452-0>) has excellent illustrations to guide

readers through their findings and methods.

Response: We thank the reviewer for the suggestion. We have included a new schematic showing the proposed mechanisms of the distributed LiIn network enhancing performance as a part of the revised Figure 5. This schematic was referenced in the new paragraph that describes mechanisms above. The new schematic, as well as the rest of the new Fig. 5 (which shows cryo-FIB characterization of pure Al and Al-In foils at different stages of cycling) is included below.

Figure 5. Cryogenic-FIB-SEM of Al_{94.5}In_{5.5} and aluminum foils at different stages of cycling in SSBs. (A) Pristine aluminum foil, (B) aluminum foil after full lithiation, and (C) aluminum foil after delithiation. (D) Pristine Al_{94.5}In_{5.5} foil, (E) Al_{94.5}In_{5.5} foil after LiIn formation, (F) Al_{94.5}In_{5.5} foil after full lithiation, and (G) Al_{94.5}In_{5.5} foil after delithiation. All cells for FIB-SEM were assembled with a stack pressure of 50 MPa and cycled at 0.7 mA cm⁻² with cathode loading of 3.0 mAh cm⁻²,

with both sets of foils being 11 μm in thickness instead of the 30 μm -thick foils used elsewhere to ensure FIB milling effectiveness. **(H)** Schematics showing reaction mechanisms in the Al-In foil with a multiphase microstructure. (i) Pristine Al-In foil in contact with SSE. (ii) The distributed indium phase is lithiated first to form LiIn. (iii) The aluminum phase begins to react with lithium transported from the LiIn phase. (iv) During delithiation, the LiIn phase remains lithiated and can transport lithium to avoid trapping by the pure aluminum phase.

Comment: The manuscript itself is well written, and the importance of the findings are also appropriate for NatComm. However, taking in mind the above comments, the reviewer suggests a Major Revision for the manuscript prior to acceptance/publication.

Response: Thanks again to the reviewer for the very thoughtful comments – clearly, they have helped us to improve the paper.

Reviewer #2 (Remarks to the Author):

In this manuscript, the authors designed a solid-state battery with Al-In foil anodes and enhanced the rate behaviors as well as reversibility. However, the logic of the article is somewhat confusing. In addition, the authors need to prove the storage mechanism of the materials through other characterization methods and theoretical calculations. Therefore, I'm afraid that this work cannot be considered for publishing on Nature Communications. Other comments are as follows:

Comment: 1. Why the incorporation of small amounts of indium within aluminum foils can greatly influence the lithium diffusion kinetics? What's the synergetic effect between indium and aluminum?

Response: This comment gets to the heart of our proposed mechanism, and in response to this comment and others from other reviewers we have revised and clarified our proposed mechanism based on our measurements. The following changes have been made:

On page 12 of the manuscript, we have removed our prior paragraphs on the extraction of lithium diffusion coefficients from the GITT data and have replaced it with the following paragraph:

“Although GITT is sometimes used to extract diffusion coefficients from electrochemical data, this was not performed here since such analysis requires single-phase reaction behavior, and both the Al and In react via two-phase reactions⁴⁴. However, prior work using nuclear magnetic resonance (NMR) techniques has directly measured Li diffusion coefficients in both the LiAl and LiIn phases⁴⁸. The Li diffusion coefficients in both phases are quite high, with that for LiIn being approximately $10^{-6} \text{ cm}^2 \text{ s}^{-1}$ and that for LiAl being approximately $10^{-7} \text{ cm}^2 \text{ s}^{-1}$ at room temperature. There is also some variation of the Li diffusion coefficient with composition in both phases since both have a narrow range of single-phase solubility (for instance, from $\text{Li}_{48.3}\text{Al}_{51.7}$ to $\text{Li}_{53.1}\text{Al}_{56.9}$; see binary phase diagrams in Supplementary Fig. 6). The high Li diffusion coefficient in LiIn is consistent with the known high rate capabilities of pure In anodes⁴⁹, and the lithiated phases of both In and Al can support relatively fast solid-state Li transport.”

On page 17 of the manuscript, we have rewritten the prior paragraph on proposed mechanisms as follows:

“Taken together, these findings provide evidence that the distributed LiIn phase within the aluminum matrix is important for enhancing the reversibility, rate behavior, and performance of Al-In electrodes. As already noted, the indium phase is lithiated first and stays lithiated even after discharge of full cells, as shown in the schematic in Fig. 5H(i-ii). This LiIn phase, which can support relatively fast Li diffusion, likely influences behavior in the following ways. First, since the LiIn phase is distributed throughout the aluminum matrix as a layered 3D network, there is a greater interfacial area available for the reaction of the aluminum with lithium from the LiIn phase. This enables transport of lithium from the LiIn network to react with aluminum to form LiAl with a lower overpotential (Fig. 5H(iii)). This idea is supported by the GITT measurements in Fig. 3D, where the Al-In electrode shows approximately ~100 mV lower overpotential during aluminum lithiation compared to the pure aluminum electrode, while both show almost the same OCV values. The improved rate behavior in Fig. 3B is also likely due to this 3D distributed network effect. Second, the distributed LiIn phase appears to play an important role in minimizing lithium trapping, which is a known failure mechanism in aluminum electrodes since the pure delithiated aluminum phase exhibits a low Li diffusion coefficient and can act as a physical barrier to further Li extraction⁵³. The distributed LiIn phase provides high-diffusivity transport channels through which Li can be removed through the surrounding delithiated aluminum phase (Fig. 5H(iv)), enabling the high initial CE observed in the Al-In cells (Fig. 2). Overall, these data show that the design concept of an interspersed mixed-ion-electron-conducting phase within a dense foil proves to be effective for improved CE and rate behavior.”

The new schematic that describes our mechanism referenced above (Fig. 5H) is included in response to another comment below.

Comment: 2.The effects of Indium content and stack pressure should be considered first. In addition, the stack pressure should be consistent in the manuscript. For example, the stack pressure in Figure 2A-D and Figure S2 is 24 MPa, but it changed to 50 MPa in Figure 2(G, H). Moreover, the assembled stack pressure of the cells for SEM in Figure 5 changed to 24 MPa again.

Response: We agree that stack pressure is an important consideration for both material evolution and battery performance in our experiments. In our manuscript, we tested the behavior of cells with Al-In anodes at different stack pressures, and that data are reported in Supplementary Figure 12, as reproduced below.

Supplementary Figure 12. Investigating of effects of stack pressure on full cells with Al_{94.5}In_{5.5} anodes with 8.3 mAh cm⁻² of NMC622 cathode loading. **(A)** Voltage curves of the first cycle under

stack pressures of 15, 24, 50, and 70 MPa using a current density of 0.8 mA cm⁻². **(B)** Voltage curves of the 10th cycle under a current density of 6.5 mA cm⁻².

As these data show, stack pressures of 24 MPa and 50 MPa result in high first-cycle capacity, with slightly lower capacity on cycle 10 with the lower stack pressure. This baseline data is important for showing behavior under different conditions. We included cycling data at different stack pressures in the manuscript to highlight that in fact 50 MPa is not required.

To further address the reviewer's question, we have added a new paragraph on page 19: "We note that the stack pressure affects all components of these cells (cathode, separator, anode, and interfaces), and that different stack pressures could possibly influence the morphological evolution of the foil itself. Additional work is needed to understand the impact of stack pressure on foil anodes and their interactions with the rest of the cell stack. Importantly, though, it is expected that SSB architectures could enable advantages for alloy foil anodes even at low stack pressures, since there will never be SEI formation in the interior of the foil as occurs at porous regions of foils in liquid cells²², leading to rapid capacity decay (Supplementary Fig. 10)."

Comment: 3.Binary phase diagrams of Li-Al and Li-In may help to get further understanding of the alloy reaction.

Response: The phase diagrams have been included as a new figure in the SI to provide additional information about the phase landscape of these alloys. Note that despite the presence of other phases on the diagrams, both the Al and the In materials are lithiated to a maximum content of one Li atom per host metal atom (LiIn and LiAl) in our study. This is because lithiation of Al to more lithiated alloys is kinetically hindered at room temperature, and further lithiation of In occurs on plateaus near the Li metal potential. The new Supplementary Figure 6 in the SI is included below:

Supplementary Figure 6. Binary phase diagrams for (A) Al and Li; (B) In and Li. Panel (A) is modified from data in supplementary ref. 1, and panel (B) is modified from data in supplementary ref. 2.

Comment: 4.In Ex situ X-ray diffraction tests, more decent steps during the lithiation process are needed to testify the alloy reaction orders of Li with Al and In.

Response: We have carried out an additional experiment in which we performed ex-situ XRD of an Al-In electrode after only the initial lithiation of the indium portion of the foil. In accord with the electrochemical data in our paper, the results showed that the aluminum is not lithiated but that the LiIn phase is present. These new data have been included in Fig. 4 in the revised manuscript, as copied below.

Figure 4: Ex situ XRD characterization of foil evolution. (A) XRD analysis of aluminum anodes in the pristine state (black), after initial charge (red), after initial discharge (blue), and after 50 cycles in the discharged state (green). **(B)** XRD analysis of $\text{Al}_{94.5}\text{In}_{5.5}$ anodes in the pristine state (black), after indium lithiation (purple), after full charge (red), after discharge (blue), and after 100 cycles in the discharged state (green). All cells for (A, B) featured 5.8 mAh cm^{-2} areal capacity in the cathode, lithiating $\sim 70\%$ of the Al foil and $\sim 64\%$ of the Al-In foil in the first charge.

The following statement was added on page 14: "After partial charge so that only the indium is lithiated (purple trace), the spectrum shows the presence of LiIn (ICDD 04-017-5865, see binary phase diagram in Supplementary Fig. 6) and aluminum."

Comment: 5. What's the volume expansion ratio of pure In foil in lithiation process? Why the presence of In metal can reduce the volume expansion of the electrode?

Response: The theoretically expected volume expansion for the lithiation of In to the LiIn phase is ~85% based on the molar volumes of the In and the LiIn phases; this value is similar to that for aluminum. As mentioned, the indium phase within the foils becomes lithiated first and stays lithiated during cycling, so it does not contribute to cyclic volume changes. The overall volume of the Al-In electrodes is reduced after cycling since we believe that the inclusion of the In phase prevents excess internal porosity (cracks and voids) from accumulating.

To address this question, we have added the following statement on page 15: “**Supplementary Fig. 7 shows similar imaging results but with manual slicing of electrodes without FIB, with the delithiated aluminum showing increased thickness and porosity.**”

Comment: 6. Compared with the previously reported works, whether the rate performance, cycling stability and diffusion coefficients of the batteries are improved?

Response: To address this comment, we have added Table S1 into the Supplementary Information, which contains information on a selection of recently published works focused on alloy anodes for solid-state batteries. This table enables readers to compare our results to other work. The table is on page 10 of the SI and is referenced on page 10 of the manuscript:

“**Table S1 compares the cycling results in Fig. 2 and 3A to other recent demonstrations of alloy anode-based SSBs.**”

Table S1 is copied below, and compares thicknesses, prelithiation, areal capacity, current density, cyclability, CE, capacity retention, and stack pressure. As is evidenced by this comparison, our materials are cycled at higher current densities, similar areal capacities, and show similar cyclabilities to other alloy anodes. The lithium metal anode from Samsung (first line) exceeds other reports in terms of cyclability. The references for this table are included below the table.

Table S1. Comparison of cycling performance of ASSBs presented in this work to other literature reports.

Anode	Type of cell	Thickness (μm)	Pre-Lithiated?	Capacity (mAh cm ⁻²)	Current Density (mA cm ⁻²)	Cycles	Avg. CE (%)	Capacity Retention (%)	Stack Pressure (MPa)	Ref.
Li/Ag-C	Full	5 - 10	No	4.6	3.4	1000	99.8	89	2	3
Porous Si	Half	4.7	No	2.2	0.1	100	99.8	93	120	4
Micro-Si	Full	12	No	2.0	5	500	99.9	80	50	5
In-Li1%	Full	50	Yes	4.0	4.8	740	99.98	92	760	6
Li _{0.8} Al	Full	-	Yes	1.3	0.35	200	99.96	93	300	7
Al	Full	100	Yes	2.2	0.72	300	100	~100	100	8
Al-In	Full	30	No	2.3	6.5	200	99.7	85	50	This work
	Half	30	No	2.1	2	500	98.9	100		

3. Lee Y., Fujiki, S., Jung, C., Suzuki, N., Yashiro, N., Omoda, R., Ko, D., Shiratsuchi, T., Sugimoto, T., Ryu, S., Hwan Ku, J., Watanabe, T., Park, Y., Aihara, Y., Im, D., & Taek Han, I. High-energy long-cycling all-solid-state lithium metal batteries enabled by silver-carbon composite anodes. *Nat. Energy* **5**, 299–308 (2020).
4. Sakabe, J., Ohta, N., Ohnishi, T., Mitsuishi, K., & Takada, K. Porous amorphous silicon film anodes for high-capacity and stable all-solid-state lithium batteries. *Commun. Chem.* **1**, 24 (2018).
5. Tan, D. H. S., Chen, Y., Yang, H., Bao, W., Sreenarayanan, B., Doux, J., Li, W., Lu, B., Ham, S., Sayahpour, B., Scharf, J., Wu, E. A., Deysher, G., Han, H. E., Hah, H. J., Jeong, H., Lee, J. B., Chen, Z., & Meng, Y. S., Carbon-free high-loading silicon anodes enabled by sulfide solid electrolytes. *Science* **373**, 1494–1499 (2021).
6. Wang, Z., Zhao, J., Zhang, X., Rong, Z., Tang, Y., Liu, X., Zhu, L., Zhang, L., & Huang, J. Tailoring lithium concentration in alloy anodes for long cycling and high areal capacity in sulfide-based all solid-state batteries. *eScience* **3**, 100087 (2022).
7. Pan, H., Zhang, M., Cheng, Z., Jiang, H., Yang, J., Wang, P., He, P., & Zhou, H. Carbon-free and binder-free Li-Al alloy anode enabling an all-solid-state Li-S battery with high energy and stability. *Sci. Adv.* **8**, eabn4372 (2022).
8. Fan, Z., Ding, B., Li, Z., Hu, B., Xu, C., Xu, C., Dou, H., & Zhang, X. Long-cycling all-solid-state batteries achieved by 2D interface between prelithiated aluminum foil anode and sulfide electrolyte. *Small* **2204037** (2022).

Comment: 7. In Figure 3B, what does the Q2 represent in the equivalent circuit? It needs to be explained.

Response: We have added a statement in the caption of Fig. 3 that explains the equivalent circuit: “The equivalent circuit features two resistor elements (R_1 and R_2) as well as a constant phase element (Q_2).”

Comment: 8. The storage mechanism is not clear. TEM characterization and other in-situ analysis need to be supplemented.

Response: We agree with the reviewer that further characterization of material evolution would be helpful for understanding charge storage characteristics. Given the length scale of the microstructure of the foils (microns), we do not believe that TEM would provide comprehensive information regarding mechanisms. Instead, we have carried out new cryo-focused ion beam (FIB)-SEM imaging of the Al-In foils after different stages of cycling to investigate microstructure evolution. This provides more useful information than the SEM cross sections that were previously shown in paper, since those samples were cut with a scalpel by hand, which obscures microstructural features. The new data are shown in the revised Fig. 5, which is included below. The previous data have been moved to the supplementary information (Supplementary Figure 7). A new author (Sun Geun Yoon) has also been added to the manuscript since he helped perform these experiments.

Figure 5. Cryogenic-FIB-SEM of Al_{94.5}In_{5.5} and aluminum foils at different stages of cycling in SSBs. (A) Pristine aluminum foil, (B) aluminum foil after full lithiation, and (C) aluminum foil after delithiation. (D) Pristine Al_{94.5}In_{5.5} foil, (E) Al_{94.5}In_{5.5} foil after LiIn formation, (F) Al_{94.5}In_{5.5} foil after full lithiation, and (G) Al_{94.5}In_{5.5} foil after delithiation. All cells for FIB-SEM were assembled with a stack pressure of 50 MPa and cycled at 0.7 mA cm⁻² with cathode loading of 3.0 mAh cm⁻², with both sets of foils being 11 μm in thickness instead of the 30 μm-thick foils used elsewhere to ensure FIB milling effectiveness. (H) Schematics showing reaction mechanisms in the Al-In foil with a multiphase microstructure. (i) Pristine Al-In foil in contact with SSE. (ii) The distributed indium phase is lithiated first to form LiIn. (iii) The aluminum phase begins to react with lithium transported from the LiIn phase. (iv) During delithiation, the LiIn phase remains lithiated and can transport lithium to avoid trapping by the pure aluminum phase.

The following section has been added to page 15:

“Figure 5 shows cryo-FIB-SEM imaging results from $\text{Al}_{94.5}\text{In}_{5.5}$ and pure aluminum electrodes at different stages of cycling. The cryogenic temperatures are useful for ensuring minimal sample damage and interaction with the Ga^+ milling beam. Figure 5A shows a pristine aluminum electrode. Figure 5B shows an aluminum electrode after full lithiation, where it has clearly grown in thickness and shows distinct cracks. Figure 5C shows an aluminum electrode after delithiation, and it contains some porosity. Supplementary Fig. 7 shows similar imaging results but with manual slicing of electrodes without FIB, with the delithiated aluminum showing increased porosity. Figure 5D shows a cryo-FIB image of a pristine $\text{Al}_{94.5}\text{In}_{5.5}$ foil, with the lighter-contrast indium phase visible as layers throughout the aluminum matrix. Figure 5E shows an $\text{Al}_{94.5}\text{In}_{5.5}$ foil electrode after lithiation of only the indium. After full lithiation of the foil (Fig. 5F), the material undergoes a $\sim 100\%$ thickness expansion, and after delithiation of the aluminum (Fig. 5G), the thickness of the foil has shrunk. Importantly, the LiIn layers remain intact and visible throughout the entire lithiation/delithiation process. A FIB-SEM image after 20 cycles (Supplementary Fig. 8) also shows intact and connected LiIn layers, indicating that the foil remains dense with interconnected LiIn layers throughout cycling. After undergoing 100 charge/discharge cycles, the foils remained mechanically intact (Supplementary Fig. 9).”

An Al-In electrode milled and imaged after 20 cycles is shown in the Supplementary Fig. 8:

Supplementary Figure 8. Cryogenic-FIB-SEM of an $\text{Al}_{94.5}\text{In}_{5.5}$ electrode after 20 full lithiation/delithiation cycles. The cell was assembled with a stack pressure of 50 MPa and cycled at 0.7 mA cm^{-2} with cathode loading of 3.0 mAh cm^{-2} , with the pristine foil being $11 \mu\text{m}$ thick. The morphology of the surface is due to curtaining during FIB milling.

We have also replaced the original milled SEM image in Fig. 1C with a new cryo-FIB image of the pristine material for consistency with these new data.

Finally, we have added the following information to the experimental section: “Cryogenic FIB-SEM images were collected on a Thermo Fisher Helios 5CX instrument using a temperature of $-145 \text{ }^\circ\text{C}$ for milling, polishing, and imaging. FIB milling of the electrodes used 30 keV Ga^+ ions at 9.4 nA and polishing was performed at 2.5 nA.”

Comment: 9. In the manuscript, the loading of the cathode (8.3 mAh/cm^2 or 5.8 mAh/cm^2) are suggested to be fixed.

Response: We used different cathode loadings to test the anodes under different extent of lithiation/delithiation. Since the main focus of our paper is on the anodes themselves, we believe this is appropriate and have elected to maintain this component of the paper as-is.

To expand the cathode loading conditions tested, we have added additional data to the manuscript in Fig. 3A that shows cycling under conditions in which a significant excess of cathode capacity is included ($\sim 16 \text{ mAh cm}^{-2}$). This was done to simulate a “half-cell” test but without using Li metal as the counter electrode, since Li metal is prone to short circuiting during cycling in SSBs. This new test shows that the Al-In electrode can cycle under capacity-limited conditions for 500 cycles, showing durability in half-cell testing. A new paragraph has been added on page 9-10:

“As an additional cycling test to examine longer-term durability, $\text{Al}_{94.5}\text{In}_{5.5}$ electrodes were cycled in cells with a significant excess of cathode material ($\sim 16 \text{ mAh cm}^{-2}$) under capacity-limited conditions to effectively simulate a “half-cell”-type test, but without using Li metal that is prone to short circuiting. This type of test minimizes the influence of any cathode degradation in the cell since there is excess cathode active material present. Figure 3A shows that this cell exhibited 500 cycles with steady capacity and no short circuiting, where lithiation areal capacity was controlled to be 2.1 mAh cm^{-2} per cycle at a current density of 2.0 mA cm^{-2} . Supplementary Fig. 3 contains the first, 100th, and 300th voltage curves from this experiment, showing consistent curve shape from the 100th to 300th cycles. Table S1 compares the cycling results in Fig. 2 and 3A to other recent demonstrations of alloy anode-based SSBs.”

The new Fig. 3A (along with the rest of Fig. 3) is shown here:

Figure 3: Cycling, rate behavior, impedance, and GITT of foil electrodes. (A) Galvanostatic testing of an $\text{Al}_{94.5}\text{In}_{5.5}$ electrode at 0.5 mA cm^{-2} for the first cycle and 2.0 mA cm^{-2} for the subsequent cycles under constant-capacity testing conditions (lithiation capacity per cycle controlled to be 2.1 mAh cm^{-2}). This cell has a significant excess of NMC at the cathode (16 mAh cm^{-2}) to mimic a half cell, and it was tested under 50 MPa stack pressure. (B) Rate testing of Al/LPSC/NMC and $\text{Al}_{94.5}\text{In}_{5.5}/\text{LPSC/NMC}$ full cells with 8.3 mAh cm^{-2} cathode loading, 50 MPa stack pressure, and current densities denoted. (C) Nyquist plots and equivalent circuit of Al/LPSC/NMC and $\text{Al}_{94.5}\text{In}_{5.5}/\text{LPSC/NMC}$ full cells with 8.3 mAh cm^{-2} cathode loading and 50 MPa stack pressure. The right panel shows a magnified view of the spectra. The equivalent circuit features two resistor elements (R_1 and R_2) as well as a constant phase element (Q_2). (D) GITT measurements of Al/LPSC/Li (red) and $\text{Al}_{94.5}\text{In}_{5.5}/\text{LPSC/Li}$ (blue) half cells with 10 MPa stack pressure. The open circles represent OCV values after the rest periods, and the solid lines are the voltage traces during current application.

The new Supplementary Figure 3 is shown here:

Supplementary Figure 3. Voltage curves from galvanostatic testing of an $\text{Al}_{94.5}\text{In}_{5.5}$ electrode at 0.5 mA cm^{-2} for the first cycle and 2.0 mA cm^{-2} for the subsequent cycles under constant-capacity testing conditions (lithiation capacity controlled to be 2.1 mAh cm^{-2}); these curves correspond to the cycling data in Fig. 3A. This cell has a significant excess of NMC at the cathode (16 mAh cm^{-2}) to simulate a half cell, and it was tested under 50 MPa stack pressure at room temperature.

Comment: 10. There are some errors that should be corrected. For example, “(Fig. 1D)” in Line 4, Page 7; “2.5 2t.% VGCF” in Page 17; etc.

Response: Thanks for finding these errors – we have changed the text to fix these.

Comment: 11. A comparison of recently reported hosts for Li storage in solid-state batteries should be provided, so that the readers can have a clear comparison between this material and other materials.

Response: We have compared our results to other recently reported alloy anode-based SSB cells in the new Table S1.

Reviewer #3 (Remarks to the Author):

Major revision: Suitable for Nature Communications after major revision.

In this work, the authors performed a study to demonstrate the benefits of solid-state architectures along with microstructure engineering of aluminum-based foils as anodes for rechargeable battery systems. Authors found that minor alloying of 5.5 at. % indium with aluminum led to enhance reversibility and improve rate behavior. Such aluminum-indium based foil anodes remained compact during lithiation and delithiation within solid state battery framework and avoided extensive SEI formation. Given the fact that the work is timely and thoroughly done, I would recommend this topic of discussion to be published. However, there are few possible sources of ambiguity (as listed below) which need to be majorly revised before the decision of acceptance.

Comment: 1. What is reason behind huge fluctuation of the CE versus cycle number for Al anode (as shown in Fig. 2F)? In contrast to Fig. 2F, Fig. 2I exhibits similar behavior for both the Al and Al-In anodes at higher current density. How do the authors justify this?

Response: The fluctuating Coulombic efficiency values greater than 100% for the pure Al anode are likely due to lithium trapping within the foil, which is a known issue for Al caused by the low Li diffusion coefficient in the delithiated Al phase. At the low current densities in Fig. 2F, these differences are very clear. There is less fluctuation at high current densities in Fig. 2J. It is notable that the low first CE for pure Al (~60% in Fig. 2F) indicates that a significant amount of Li is left in the material, and this Li can be extracted in later cycles to result in CE>100%. To address this point, we have added the following text on page 7: "The cell with the pure aluminum anode showed more erratic CE values with some over 100%, which is likely a result of trapped lithium within the material in the first few cycles¹⁴."

Comment: 2. What is the origin of the drop (minimum at the cycle number 2) in the CE versus cycle plot, which is followed by a gradual increase, as shown Fig. 2J (inset)?

Response: The drop in the CE on cycle 2 in the inset of Fig. 2J occurred because the current density was increased from 0.8 mA cm⁻² to 6.5 mA cm⁻² at this cycle. To address this comment, we have added the following to the Fig. 2 caption: "(J) CE with cycling, with inset showing CE over the first 10 cycles; the drop in CE on cycle 2 in the inset is due to the increased current density during this cycle."

Comment: 3. Line no. 211-212: What do the author mean by interfacial resistance? Do they mean the anode-electrolyte interface or the Al-In interface? Authors should clarify this point.

Response: We have changed the sentence on page 10 to clarify this point: "These data suggest that the presence of indium reduces the interfacial resistance of the anode/SSE interface."

Comment: 4. High order of Li-diffusivity for Al-In alloy anode is only observed at a controlled and low value of areal capacity (0.5 mAh cm⁻²), as shown in Fig. 3D. In addition, the data points

are fluctuating. Especially, the Li-diffusion behavior of both the Al and Al-In anodes become similar at the Al-plateau region. Hence, authors should explain the origin of the improved rate performance for Al-In anode with deeper insight.

Response: In response to this comment and others from other reviewers, we have revised and clarified our proposed mechanism. We have also removed the calculation of diffusion coefficients, as our previous calculations were based on the inaccurate assumption of a single phase reaction in these materials. The following changes have been made:

On page 12 of the manuscript, we have removed our prior paragraphs on the extraction of lithium diffusion coefficients from the GITT data and have replaced it with the following paragraph:

“Although GITT is sometimes used to extract diffusion coefficients from electrochemical data, this was not performed here since such analysis requires single-phase reaction behavior, and both the Al and In react via two-phase reactions⁴⁴. However, prior work using nuclear magnetic resonance (NMR) techniques has directly measured Li diffusion coefficients in both the LiAl and LiIn phases⁴⁸. The Li diffusion coefficients in both phases are quite high, with that for LiIn being approximately 10^{-6} cm² s⁻¹ and that for LiAl being approximately 10^{-7} cm² s⁻¹ at room temperature. There is also some variation of the Li diffusion coefficient with composition in both phases since both have a narrow range of single-phase solubility (for instance, from Li_{48.3}Al_{51.7} to Li_{53.1}Al_{56.9}; see binary phase diagrams in Supplementary Fig. 6). The high Li diffusion coefficient in LiIn is consistent with the known high rate capabilities of pure In anodes⁴⁹, and the lithiated phases of both In and Al can support relatively fast solid-state Li transport.”

On page 17 of the manuscript, we have rewritten the prior paragraph on proposed mechanisms as follows:

“Taken together, these findings provide evidence that the distributed LiIn phase within the aluminum matrix is important for enhancing the reversibility, rate behavior, and performance of Al-In electrodes. As already noted, the indium phase is lithiated first and stays lithiated even after discharge of full cells, as shown in the schematic in Fig. 5H(i-ii). This LiIn phase, which can support relatively fast Li diffusion, likely influences behavior in the following ways. First, since the LiIn phase is distributed throughout the aluminum matrix as a layered 3D network, there is a greater interfacial area available for the reaction of the aluminum with lithium from the LiIn phase. This enables transport of lithium from the LiIn network to react with aluminum to form LiAl with a lower overpotential (Fig. 5H(iii)). This idea is supported by the GITT measurements in Fig. 3D, where the Al-In electrode shows approximately ~100 mV lower overpotential during aluminum lithiation compared to the pure aluminum electrode, while both show almost the same OCV values. The improved rate behavior in Fig. 3B is also likely due to this 3D distributed network effect. Second, the distributed LiIn phase appears to play an important role in minimizing lithium trapping, which is a known failure mechanism in aluminum electrodes since the pure delithiated aluminum phase exhibits a low Li diffusion coefficient and can act as a physical barrier to further Li extraction⁵⁰. The distributed LiIn phase provides high-diffusivity transport channels through which Li can be removed through the surrounding delithiated aluminum phase (Fig. 5H(iv)), enabling the high initial CE observed in the Al-In cells (Fig. 2). Overall, these data show that the design concept of an interspersed mixed-ion-electron-conducting phase within a dense foil proves to be effective for improved CE and rate behavior.”

The new schematic that describes our mechanism is included in the revised Fig. 5, as copied below:

Figure 5. Cryogenic-FIB-SEM of $\text{Al}_{94.5}\text{In}_{5.5}$ and aluminum foils at different stages of cycling in SSBs. (A) Pristine aluminum foil, (B) aluminum foil after full lithiation, and (C) aluminum foil after delithiation. (D) Pristine $\text{Al}_{94.5}\text{In}_{5.5}$ foil, (E) $\text{Al}_{94.5}\text{In}_{5.5}$ foil after LiIn formation, (F) $\text{Al}_{94.5}\text{In}_{5.5}$ foil after full lithiation, and (G) $\text{Al}_{94.5}\text{In}_{5.5}$ foil after delithiation. All cells for FIB-SEM were assembled with a stack pressure of 50 MPa and cycled at 0.7 mA cm^{-2} with cathode loading of 3.0 mAh cm^{-2} , with both sets of foils being $11 \mu\text{m}$ in thickness instead of the $30 \mu\text{m}$ -thick foils used elsewhere, to ensure FIB milling effectiveness. (H) Schematics showing reaction mechanisms in the Al-In foil with a multiphase microstructure. (i) Pristine Al-In foil in contact with SSE. (ii) The distributed indium phase is lithiated first to form LiIn. (iii) The aluminum phase begins to react with lithium transported from the LiIn phase. (iv) During delithiation, the LiIn phase remains lithiated and can transport lithium to avoid trapping by the pure aluminum phase.

Comment: 5. A minor comment: (line no. 147) “Differential capacity curves comparing the first cycle of two cells with different anodes (Fig. 1D) highlight the improved reversibility of the Al-In-based cell.” Numbering of figure is inappropriate.

Response: Thanks for catching this error – we have fixed it in the revised manuscript.

REVIEWERS' COMMENTS

Reviewer #1 (Remarks to the Author):

The authors have reexamined their previous submission and considered new information regarding ion diffusivity/transport, interfaces, etc. This, combined with the added characterization (e.g. Fig 5), has made the explanation of the results more compelling and plausible. There are some lingering questions, but these are beyond the scope of this work and unlikely to be resolved in the near term.

Nevertheless, this article can be merited for bringing such issues forward into the public domain for others to follow and build upon. Particularly with respect to Li transport in multiphase alloy systems where expansion/contraction, interfacial integrity, and electrochemical potentials make for an exciting and important area for ASSB development since it is clear that indium alone is ill-suited as a LIB anode material.

Given that the methods and materials used are both well described in the text, I recommend publication in Nature Communications.

Reviewer #3 (Remarks to the Author):

Authors have responded to all the queries appropriately. Hence, I would recommend this work to be published in Nature Communications.

Response to Reviewer Comments

“Aluminum Foil Negative Electrodes with Multiphase Microstructure for All-Solid-State Li-Ion Batteries”

Yuhgene Liu¹, Congcheng Wang², Sun Geun Yoon², Sang Yun Han², John A. Lewis¹, Dhruv Prakash¹, Emily J. Klein¹, Timothy Chen², Dae Hoon Kang³, Diptarka Majumdar³, Rajesh Gopaldaswamy³, Matthew T. McDowell^{1,2*}

We thank the reviewers for their time and effort in reviewing our manuscript. In this response document, we list the reviewers' comments and describe the changes that have been made to the manuscript and SI in response to the comments. For clarity, the reviewer's comments are in black text, and our responses are in blue text. Additions and revisions to the manuscript and SI have been included in this response, and they are given in red text. Copies of the manuscript and SI in which changes are highlighted have also been included with this submission.

REVIEWER COMMENTS

Reviewer #1 (Remarks to the Author):

Comment: The authors have reexamined their previous submission and considered new information regarding ion diffusivity/transport, interfaces, etc. This, combined with the added characterization (e.g. Fig 5), has made the explanation of the results more compelling and plausible. There are some lingering questions, but these are beyond the scope of this work and unlikely to be resolved in the near term.

Nevertheless, this article can be merited for bringing such issues forward into the public domain for others to follow and build upon. Particularly with respect to Li transport in multiphase alloy systems where expansion/contraction, interfacial integrity, and electrochemical potentials make for an exciting and important area for ASSB development since it is clear that indium alone is ill-suited as a LIB anode material.

Given that the methods and materials used are both well described in the text, I recommend publication in Nature Communications.

Response: We thank the reviewer for their careful reading of the manuscript and helpful comments.

Reviewer #3 (Remarks to the Author):

Comment: Authors have responded to all the queries appropriately. Hence, I would recommend this work to be published in Nature Communications.

Response: We thank the reviewer for their suggestion to publish.